# The Hox transcription factor Ubx stabilizes lineage commitment by suppressing cellular plasticity in *Drosophila*

**Katrin Domsch[1], Julie Carnesecchi[1], Vanessa Disela[1], Jana Friedrich[1], Nils Trost[1], Olga Ermakova[1], Maria Polychronidou[2], Ingrid Lohmann[1]\***

[1]Centre for Organismal Studies (COS) Heidelberg, Heidelberg, Germany; [2]EMBO, Heidelberg, Germany

**Abstract** During development cells become restricted in their differentiation potential by repressing alternative cell fates, and the Polycomb complex plays a crucial role in this process. However, how alternative fate genes are lineage-specifically silenced is unclear. We studied Ultrabithorax (Ubx), a multi-lineage transcription factor of the Hox class, in two tissue lineages using sorted nuclei and interfered with Ubx in mesodermal cells. We find that depletion of Ubx leads to the de-repression of genes normally expressed in other lineages. Ubx silences expression of alternative fate genes by retaining the Polycomb Group protein Pleiohomeotic at Ubx targeted genomic regions, thereby stabilizing repressive chromatin marks in a lineage-dependent manner. Our study demonstrates that Ubx stabilizes lineage choice by suppressing the multipotency encoded in the genome via its interaction with Pho. This mechanism may explain why the Hox code is maintained throughout the lifecycle, since it could set a block to transdifferentiation in adult cells.
DOI: https://doi.org/10.7554/eLife.42675.001

**\*For correspondence:**
ingrid.lohmann@cos.uni-heidelberg.de

**Competing interests:** The authors declare that no competing interests exist.

## Introduction

Multicellular animals owe their complexity to their capacity to produce and maintain different lineages composed of multiple cell types that share virtually the same genomic DNA. Such complexity requires a tight regulation of gene expression to unambiguously specify and constrain the developmental path taken by cells within and between different lineages. Cell fates are controlled by networks of transcription factors (TFs) that act in the context of the genomic chromatin state to activate transcriptional programs realizing the distinct properties of cells of a given lineage (*Hobert, 2011*; *Levine and Davidson, 2005*; *Spitz and Furlong, 2012*). The forced expression of TFs expressed only in one lineage like Myoblast Determination Protein (MyoD), which is sufficient to alter cell identity (*Tapscott et al., 1988*; *Weintraub et al., 1989*), showed that lineage-restricted TFs can act as major lineage switches by activating lineage-specific transcriptional programs. However, recent profiling experiments revealed that global gene expression and chromatin accessibility changes are imperfect even in MyoD reprogrammed cells, showing that these highly potent reprogramming TFs cannot completely erase the original cell or lineage fate and unequivocally induce the new one (*Manandhar et al., 2017*). In addition, lineage choice does not only involve the activation of one lineage-specific transcriptional program but also the repression of the programs of all alternative lineages (*Kutejova et al., 2016*; *Mall et al., 2017*), a regulatory wiring that needs to be faithfully accomplished in all the different lineages. Although lineage-restricted TFs could in principle fulfil this function, it would ask for a highly complex regulatory architecture. On the other hand, TFs expressed in multiple/all lineages with a rather promiscuous binding behaviour would dramatically

reduce this complexity and would elegantly explain lineage-specific gene regulation based on the interaction of these broadly expressed TFs with lineage-restricted factors.

Hox TFs represent an excellent model to address this fundamental question, since they are active in all lineages along the anterior-posterior (AP) axis of bilaterian animals. Importantly, the famous and well-known identity switch of whole body parts, the homeotic transformation, which is induced by altered Hox expression (*Lewis, 1978*; *Schneuwly et al., 1987*), shows that Hox TFs have comparable functions in all lineages during development and indicates that they control the development of different lineages in a highly specific manner. The latter conclusion is supported by recent experiments showing that mesoderm-derived vascular wall mesenchymal stem cells (VW-MSCs) can be generated in vitro from induced pluripotent stem cells (iPSCs) simply by inducing the expression of a mixture of *Hox* genes that are selectively expressed in adult VM-MSCs (*Steens et al., 2017*). However, while this study showed that *Hox* genes alone are sufficient to induce the generation of one specific cell type of one lineage in vitro, it left the questions open whether this is also the case in vivo, and how Hox TFs unambiguously select among the many possible transcriptional programs only one, which drives a cell or a whole lineage into one specific direction.

One major bottleneck in this direction is that genome-wide Hox chromatin binding studies have been performed so far mainly in cell culture systems (*Beh et al., 2016*; *Zouaz et al., 2017*), specialized epithelial tissues (*Agrawal et al., 2011*), or mixtures of cell lineages (*Shlyueva et al., 2016*), hampering the identification of common and lineage-specific mechanisms employed by Hox TFs in different lineages in vivo. Furthermore, unlike lineage-restricted TFs, which are often tested in vivo using ectopic expression systems (*Fukushige and Krause, 2005*; *Patel and Hobert, 2017*), the functional analysis of TFs acting in multiple lineages requires the targeted interference with these factors in individual lineages. With the availability of conditional genome editing (*Port and Bullock, 2016*; *Port et al., 2014*) and nanobody driven protein degradation systems (*Caussinus et al., 2011*) this is now possible in an efficient manner and allows to elucidate the mode of action of Hox TFs in individual lineages, which are located in an otherwise unperturbed tissue environment at any stage in development. This is particularly important for multi-lineage TFs like the Hox proteins, which extensively control cell communication (*Cunningham and Duester, 2015*; *Pearson et al., 2005*) and thus influence their own action in neighbouring lineages.

Here, we probe the Hox TF Ubx in the mesodermal and neuronal lineages using sorted nuclei of *Drosophila* embryos and by interfering with Ubx function specifically in the mesoderm lineage that is specified and fully committed to the mesodermal fate. To this end, we generated a GFP-Ubx gene fusion at the endogenous locus using CRISPR-Cas9 and homologous recombination. We show that Ubx is a key regulator of lineage development and diversification, as it controls the mesodermal and neuronal transcriptional programs with high specificity despite interacting with many genes in both lineages. Intriguingly, our study demonstrates that Ubx controls lineage specification and differentiation not primarily by activating lineage-specific gene programs but to a large extent by restricting cellular plasticity within each lineage. In the mesoderm Ubx executes this function by silencing alternative fate genes, and we show that this repression requires the interaction of Ubx with the Polycomb Group (PcG) protein Pleiohomeotic (Pho) on co-bound chromatin sites. We furthermore find that Ubx stabilizes Pho binding to Ubx targeted genomic regions, and that this interaction is critical for gene repression by controlling the balance of H3K27me3 as well as H3K27ac at these sites. Taken together, our study demonstrates that Hox TFs control not only the segmental identity but also the developmental programs intrinsic to each lineage with high precision, and that one of the prominent functions of this multi-lineage TF family is the repression of alternative lineage programs, which restricts the cellular potential in a lineage-specific manner.

## Results

### A comprehensive map of transcriptional profiles and Ubx binding in the *Drosophila* embryonic mesodermal and neuronal lineages

To elucidate mechanisms that enable a multi-lineage TF to instruct the development of divergent lineages, we recorded on the tissue level chromatin binding of the broadly expressed Hox protein Ubx in the mesoderm and nervous system and determined the transcriptomes of these two lineages during two stages of *Drosophila* embryogenesis. To this end, we used the isolation of nuclei tagged in

specific cell types (INTACT) method (*Steiner et al., 2012*) by inducing the expression of the nuclear membrane protein Ran GTPase activating protein (RanGAP) fused to a biotin ligase acceptor peptide and the *E. coli* Biotin ligase (BirA) in the mesoderm via the pan-mesodermal *twist* (*twi*) (*Leptin and Grunewald, 1990*) or the pan-neuronal *embryonic lethal abnormal vision* (*elav*) (*Robinow and White, 1991*; *Yao and White, 1994*) regulatory regions. Co-expression of both transgenes in *Drosophila* embryos, which did not affect development (*Figure 1—figure supplement 1C– 1H*), resulted in cell type-specific biotinylation of nuclei in vivo (*Figure 1A–C'*). Mesodermal or neuronal nuclei were efficiently isolated using Streptavidin-coated beads, as we detected Mef2-positive nuclei and concomitantly mesodermal marker gene expression like *Mef2 and nautilus* (*nau*) almost exclusively in the mesodermal collection (*Figure 1D*), while Elav-labelled neuronal nuclei and

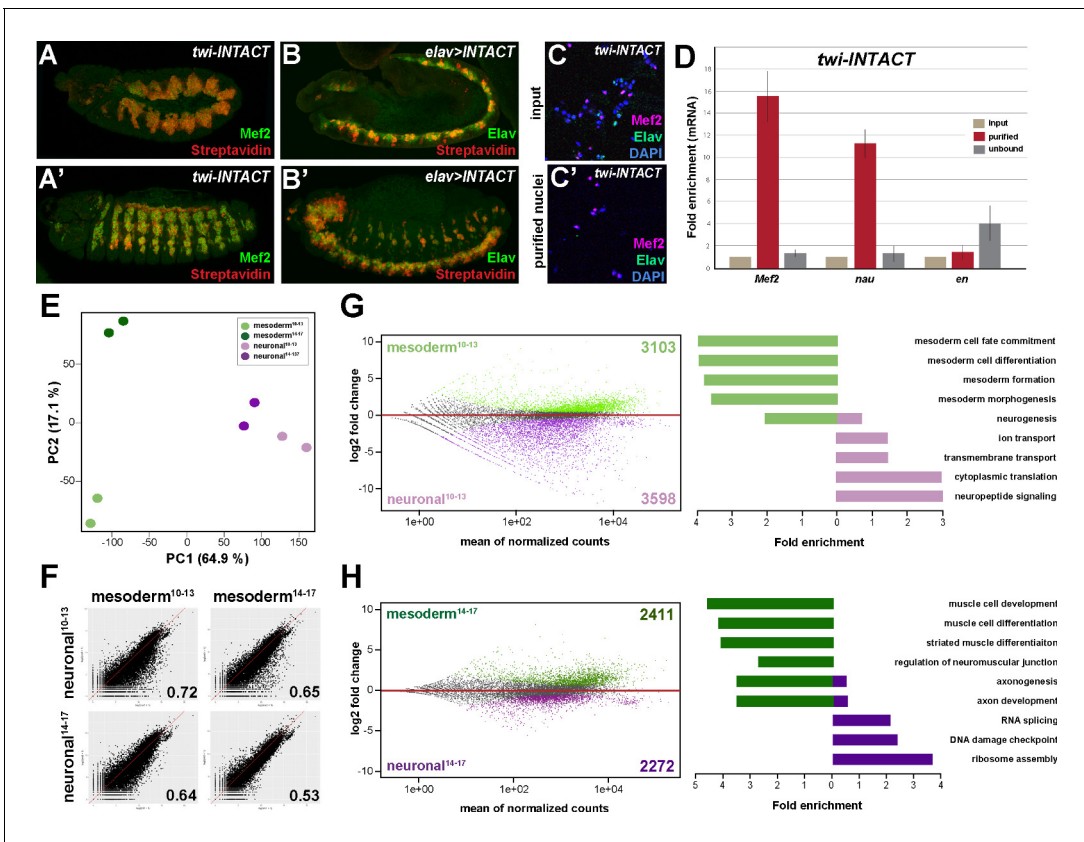

**Figure 1.** Comparative profiling of the mesodermal and neuronal transcriptomes in *Drosophila* embryos. (A, A') Lateral views of stage 11 (A) and stage 14 (A') *twi-INTACT Drosophila* embryos stained for the muscle differentiation marker Mef2 (green) and Streptavidin (red). (B, B') Lateral view of stage 11 (B) and stage 14 (B') *elav >INTACT Drosophila* embryos stained for the pan-neuronal marker Elav (green) and Streptavidin (red). (C, C') Nuclei obtained from *twi-INTACT* embryos stained for Mef2 (red), Elav (green) and DAPI (blue) before (C) and after (C') INTACT purification. (D) Expression of mesodermal (*Mef2*, *nau*) and ectodermal (*en*) mRNA transcripts in nuclei purified from *twi*-INTACT embryos as measured by RT-PCR using RNA from total/input (10%), purified and unbound nuclei. (E) Principle Component Analysis (PCA) applied to all RNA-seq samples identifies three clusters: one corresponding to mesodermal specification events (mesoderm[10-13]), one to mesodermal differentiation events (mesoderm[14-17]) and the neuronal cluster sharing similar expression signatures. The two mesodermal cluster are distinct from the neuronal and result in the mesodermal group. (F) Pearson correlation analysis highlights the differences and similarities between the mesodermal and neuronal RNA-seq datasets. (G, H) Left: MA analysis identifies genes differentially expressed in the mesodermal and neuronal lineages at two different time windows, embryonic stage 10–13 (G) and embryonic stage 14–17 (H). Genes differentially expressed in the mesodermal lineage are indicated as green dots, those in the neuronal lineage as purple dots. Right: Bar diagram displaying fold enrichment of gene ontology terms of genes differentially expressed in the mesodermal or neuronal lineages at the two different time windows.

DOI: https://doi.org/10.7554/eLife.42675.002

The following figure supplement is available for figure 1:

**Figure supplement 1.** Comparative profiling of the mesodermal and neuronal transcriptomes in *Drosophila* embryos.

DOI: https://doi.org/10.7554/eLife.42675.003

neuronal marker genes like *eagle* (*eg*) and *elav* were found highly enriched in the neuronal fraction only (*Figure 1—figure supplement 1A–B*).

We analysed the nuclear transcriptome by high-throughput RNA sequencing (RNA-seq) using INTACT-sorted mesodermal or neuronal nuclei obtained from stage 10 to 13 (4–9 hr after egg lay (AEL)) and stage 14 to 17 embryos (10–18 hr AEL). Robust and reproducible data were obtained for all samples in biological duplicates. In total, transcripts (containing >5 RPKM) corresponding to 5652 coding genes were identified for the early mesodermal, 5716 coding genes for late mesodermal, 6752 coding genes for the early neuronal and 6182 coding genes for the late neuronal nuclei populations, which included genes typical for the lineage type and developmental stage (*Supplementary file 1*). Pearson correlation coefficient analysis revealed that the transcriptome of the mesodermal lineage was clearly distinct from the neuronal one at both time points (r = 0.72 for stages[10-13]; r = 0.53 for stages[14-17]) (*Figure 1F*, *Figure 1—figure supplement 1I*). This was also reflected in a high number of genes differentially expressed in the mesodermal as well as the neuronal lineages when comparing identical stages (*Figure 1G,H*). Importantly, GO term classification revealed a significant enrichment of processes typical for the respective lineage (mesoderm both stages: p-value<2.2e-16, neuronal stages[10-13]: p-value<1.26e-6, neuronal stages[14-17]: p-value<5.3e-5) (*Figure 1G and H*). In addition to elucidating differences in tissue profiles, Pearson correlation coefficient analysis also showed that global gene expression in the mesodermal lineage changed substantially over the selected time points (r = 0.78 for mesoderm stages[10-13] + stages[14-17]) (*Figure 1F and H*, *Figure 1—figure supplement 1I*). Tissue- and stage-dependent differences and similarities were very well reflected in the distances calculated by principal component analysis (PCA) (*Figure 1E*). PCA analysis also showed that in contrast to the mesoderm the neuronal transcriptomes were very similar at both developmental time frames (*Figure 1—figure supplement 1I*), which we assumed to be a consequence of the earlier onset of nervous system differentiation (*Bate, 2009*; *Hartenstein, 1993*).

We next profiled genome-wide Ubx binding in the same lineages and identical time windows by chromatin immunoprecipitation coupled to massively parallel sequencing (ChIP-seq) using $1 \times 10^6$ INTACT-sorted mesodermal and neuronal nuclei and an Ubx specific antibody generated and verified in the lab (see Materials and Methods). The sequencing was performed in biological duplicates and Pearson correlation coefficient analysis revealed a high similarity of the samples (r = 0.94–0.96). The data were benchmarked by the identification of Ubx binding to known target loci, including the well-characterized interaction of Ubx with the *decapentaplegic* (*dpp*) (*Capovilla and Botas, 1998*; *Manak et al., 1994*) (*Figure 2A*) and *β-Tubulin at 60D* (*βTub60D*) (*Kremser et al., 1999*) (*Figure 4—figure supplement 1A*) enhancers. We found Ubx to interact with both enhancers using chromatin from INTACT sorted mesodermal but not neuronal nuclei (*Figure 2A*, *Figure 2—figure supplement 1A*), showing that our data reflected Ubx interactions in vivo. However, this analysis also uncovered that Ubx, which is known to have different functions in the developing mesoderm and nervous system (*Enriquez et al., 2010*; *Hessinger et al., 2017*; *McCormick et al., 1995*; *Miller et al., 2001*) and which interacted with different genomic regions in the two tissue lineages (*Figure 2—figure supplement 2A*), targeted many genes in both tissue lineages (*Figure 2—figure supplement 2A*). To resolve this discrepancy, we overlapped Ubx binding events and transcriptome profiles. We found that only 27% to 36% of the Ubx chromatin interactions occurred in the vicinity of genes actively transcribed either in the mesodermal or neuronal lineages at the different stages (*Figure 2B*, *Figure 2—figure supplement 1B*), with more than 60% of these interactions occurring at intron, intergenic and distal enhancer regions (*Figure 2E*). This included *Mef2*, *teashirt* (*tsh*) and *βTub60D* in early and *tin* (*tin*), *α-Actinin* (*Actn*) and *Tropomyosin 1* (*Tm1*) in late mesodermal nuclei, while in early neuronal nuclei *deadpan* (*dpn*), *huckebein* (*hkb*) and *Neurotrophin 1* (*NT1*) in late neuronal nuclei *Neuroglian* (*Nrg*), *target of PoxN* (*tap*) and *castor* (*cas*) was among the Ubx bound active genes. In contrast, a large portion of the Ubx chromatin interactions (64% to 73%) were close to inactive genes in the two lineages (*Figure 2B*, *Figure 2—figure supplement 1B*) While it is well documented that Hox TFs function as activators and repressors depending on the context (*Pinsonneault et al., 1997*; *Saleh et al., 2000*), the high number of Ubx chromatin interactions at non-transcribed genes was unexpected. By determining the gene functions associated with Ubx interactions, we found a substantial overrepresentation of GO terms characteristic for the respective lineage among the Ubx targeted and expressed genes (mesoderm: p-value 2.2e-16, neuronal stages[10-13]: p-value 2.2e-16, neuronal stages[14-17]: p-value 1.004e-9) (*Figure 2C,D*), while Ubx

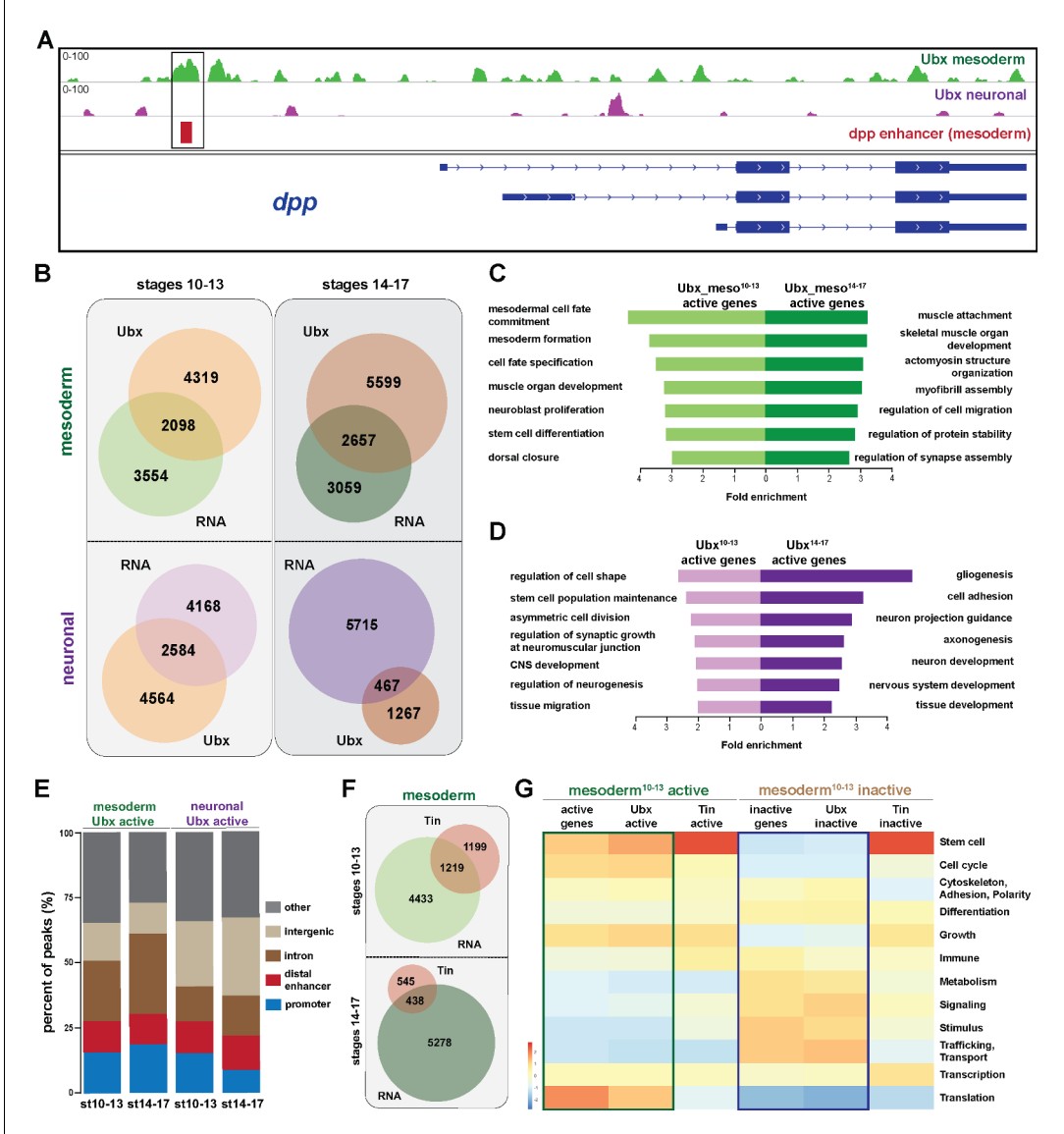

**Figure 2.** Ubx comprehensively controls tissue-specific transcriptional programs. (**A**) ChIP-seq binding profiles of Ubx at the *dpp* genomic locus in mesodermal (green) and neuronal (purple) nuclei. The different isoforms of the *dpp* gene are shown in blue, the known *dpp* visceral enhancer in red. The box highlights Ubx binding to the *dpp* visceral enhancer in mesodermal but not in neuronal cells. (**B**) Venn diagrams representing the overlaps between the mesodermal (green) and neuronal (purple) transcriptomes and Ubx bound genes (orange) at two different stages. (**C, D**) Fold enrichment of gene ontology terms of genes expressed and targeted by Ubx in the mesoderm (**C**) or the neuronal (**D**) tissues, respectively. (**E**) Comparison of the localisation of peaks called within unique genomic regions of Ubx. Locations are classified as promoters (−1000- + 10 bp from TSS, 5' UTR), distal enhancers (−2000 to −1000 from TSS, 3' UTR, downstream), intron (intronic regions), intergenic (distal intergenic) and other regions (including exons). (**F**) Venn diagrams representing the overlaps between the mesodermal (green) transcriptomes and Tin bound genes (red) at two different stages. (**G**) Heat-map displaying presence of genes belonging to higher-order categories in the different gene classes. The colour range corresponds to the centred and scaled (per column) fraction of genes annotated to the category that also appear in the sample: red colour represents high values, blue colours low fractions of genes in the category, which are also present in the sample. Rows and columns are hierarchically clustered using Euclidean distance with complete linkage. The blue and green boxes highlight the distributions of functional terms among the expressed and Ubx targeted as well as the inactive and Ubx targeted genes.

DOI: https://doi.org/10.7554/eLife.42675.004

The following figure supplements are available for figure 2:

**Figure supplement 1.** Ubx comprehensively controls tissue-specific transcriptional programs.
DOI: https://doi.org/10.7554/eLife.42675.005
**Figure supplement 2.** Ubx is a general regulator of mesoderm development.
DOI: https://doi.org/10.7554/eLife.42675.006

interactions at inactive genes occurred frequently at genes controlling processes active in other lineages (*Figure 2—figure supplements 1B–2D*). For example, ectodermal and neuronal but not mesodermal GO terms (p-value 0.7118) were highly enriched among the inactive genes bound by Ubx in the early mesodermal lineage (neuronal GO-terms: p-value 0.0008442, ectoderm GO terms: p-value 0.01196).

In order to comprehensively analyse the binding behaviour of Ubx to active and inactive genes, we used the WEADE tool, which identifies and visualizes overrepresentation of functionally related biological GO terms assembled to higher-order GO term sets and allows the representation and comparative analysis of multiple gene sets (*Trost et al., 2018*). This analysis uncovered a high correlation between the lineage-specific transcriptional profiles and the genome-wide Ubx interactions (*Figure 2G*). For example, many genes expressed in the early mesoderm control stem cell, cell cycle and translation related processes, and Ubx interactions were found enriched in the vicinity of these genes, while Ubx hardly interacted with genes encoding gene functions that were not represented among the active genes, like metabolism, signalling, stimulus and transport/trafficking related functions (*Figure 2G*). A similar correlation was found among the inactive genes, as Ubx interactions were again highly enriched at gene classes found to be overrepresented among the non-expressed genes, while Ubx did not interact with underrepresented gene classes (*Figure 2G*). This result suggested that Ubx binding controlled global gene expression in the mesoderm, the activation as well as repression, in a comprehensive manner. However, as this GO term enrichment could also be a consequence of the high number of genes bound by Ubx and not of binding preferences, we performed two types of analyses. First, we mapped the Ubx sequencing reads retrieved from stage 10 to 13 mesodermal nuclei to a previous version of the *Drosophila* genome (dm3). This resulted as expected in the identification of substantially fewer binding events (*Figure 2—figure supplement 2B*), however the genes targeted by Ubx had still a similar distribution of GO term enrichments when compared to the transcriptome according to WEADE analysis (*Figure 2—figure supplement 2C*). And second, we asked whether this binding behaviour was a general characteristic of TFs or specific for Ubx. Thus, we analysed GO term enrichment of genes bound by two mesoderm-specific TFs, Tinman (Tin), a NK homeodomain TF required for cardiac development (*Bodmer et al., 1990*) (*Azpiazu and Frasch, 1993*), and Myocyte enhancer factor 2 (Mef2), a MADS-box TF generally controlling muscle development (*Bour et al., 1995*) (*Ranganayakulu et al., 1998*). We used genome-wide binding data of Tin and Mef2 profiled at similar embryonic stages which were mapped to the dm3 *Drosophila* genome (*Jin et al., 2013*; *Sandmann et al., 2006*), and which showed that Ubx, Tin and Mef2 bound comparable numbers of chromatin sites (*Figure 2—figure supplement 2B*). WEADE analysis of genes bound by these two TFs revealed that Tin interactions did not reflect very well the gene classes represented among the active and inactive genes, while Ubx and Mef2 profiles were rather similar (*Figure 2G*, *Figure 2—figure supplement 2B,C*). Indeed, Tin preferentially interacted with genes controlling stem cell processes irrespective of whether these genes were actively transcribed or silent (*Figure 2G*, *Figure 2—figure supplement 2C*). Thus, these results strengthened our hypothesis that Ubx generally controls mesoderm development like Mef2, while Tin has a more restricted function in this tissue lineage. This is in line with Tin's published role in the determination and specification of the cardiac, visceral and dorsal mesoderm (*Azpiazu and Frasch, 1993*; *Klinedinst and Bodmer, 2003*).

In sum, these results illustrated that the broadly expressed Hox TF Ubx played a prominent role in orchestrating the transcriptional program in the mesodermal (and neuronal) lineages. In addition, our analysis revealed that a substantial fraction of Ubx interactions were found in the vicinity of inactive genes, which encoded many mesoderm-unrelated functions. Thus, we hypothesized that one important function of Ubx in tissue development could be the repression of alternative transcriptional programs, which instruct the development of other lineages.

## The generic Hox transcription factor Ubx functions as a major regulator of lineage programs

Our analysis of Ubx function in the mesodermal and neuronal lineages was so far based on correlating lineage-specific Ubx binding profiles with RNA-seq probed gene expression. In order to elucidate which genes are under direct Ubx control, we profiled the transcriptional output induced in mesodermal cells devoid of Ubx protein, while leaving Ubx levels in all other cell and tissue types unchanged. We focused our analysis on the mesoderm, as its control by various lineage-restricted

TFs is well described (*Azpiazu and Frasch, 1993*; *Klinedinst and Bodmer, 2003*; *Liu et al., 2009*; *Sandmann et al., 2006*; *Sandmann et al., 2007*; *Zaffran et al., 2002*). In order to deplete Ubx in the mesoderm, we used the targeted degradation of GFP fusion proteins. To this end, we first generated an endogenously GFP tagged version of the *Ubx* gene using the Clustered Regularly Interspaced Short Palindromic Repeats-associated Cas9 (CRISPR/Cas9)-mediated genome engineering (*Figure 3—figure supplement 1A,B*) (*Bassett et al., 2013*; *Gratz et al., 2013*) (see Materials and Methods for details). Intactness of the fusion protein in the homozygous viable GFP-Ubx fly line was verified by rescuing *Ubx* mutants using the *GFP-Ubx* allele (*Figure 3—figure supplement 1H–L*). Co-localization of GFP and Ubx protein expression in GFP-Ubx embryos (*Figure 3A,B*, *Figure 3—figure supplement 1C,D*) confirmed a precise expression control of the fusion protein. In order to degrade the GFP-Ubx fusion protein in a lineage-specific manner, we used the deGradFP system, which harnesses the ubiquitin-proteasome pathway to achieve direct depletion of GFP-tagged proteins. We first functionally verified the system by ubiquitously degrading the GFP-Ubx fusion protein using the *armadillo* (*arm*)-GAL4 driver, which resulted in a strong reduction of GFP-Ubx protein levels (*Figure 3—figure supplement 1E–G,O*), and in animals resembling the *Ubx* null mutant phenotype (*Figure 3—figure supplement 1M,N*) (*Lewis, 1978*). In a next step, we applied deGradFP to specifically interfere with Ubx function in the mesoderm using the *Mef2*-GAL4 driver (*Zars et al., 2000*). This combination substantially decreased GFP-Ubx protein accumulation in the mesoderm (*Figure 3C,D,G*), and the expression of the direct Ubx target gene *dpp* in the visceral mesoderm (*Figure 3H–J*) (*Manak et al., 1994*; *Zaffran et al., 2001*). Consequently, we only observed the well-described loss of the third midgut constriction in these embryos (*Figure 3E,F*), which is caused by the absence of Ubx activity in this tissue (*Bienz et al., 1988*), while all non-mesodermal tissues were unaffected (*Figure 3E–G*). Having confirmed the suitability of the approach, we INTACT-sorted mesodermal nuclei, in which the GFP-Ubx protein had been depleted and profiled their transcriptome using RNA-seq. Due to the activity of the *Mef2*-GAL4 driver starting at embryonic stage 7 (*Zars et al., 2000*), which resulted in significant reduction of Ubx protein levels in the mesoderm only at stage 12 (*Figure 3C,D*), we restricted our analysis to the late developmental time window (embryonic stages 14 to 17). RNA-seq experiments where performed in biological triplicates and the two most similar duplicates identified by Person correlation were used for further analysis. Comparison of the Ubx depleted and control mesodermal transcriptomes revealed a differential expression of 2845 genes, with 1393 genes showing increased mRNA accumulation, including *Fasciclin 2* (*Fas2*), *ventral veins lacking* (*vvl*), *Neurotrophin 1* (*NT1*) and *myospheroid* (*mys*), while 1452 genes exhibited reduced expression, including *βTub60D*, *Ankyrin 2* (*Ank2*) and *Notchless* (*Nle*) (*Figure 3K*). PCA analysis confirmed that the mesodermal transcriptomes in the absence (mesoderm[14-17] Ubx[Degrad]) or presence (mesoderm[14-17]) of Ubx were substantially different (*Figure 3N*). In total, 55% (791/1452) of the genes with reduced and 70% (978/1393) of the genes with increased expression were bound by Ubx in mesodermal nuclei in wild-type embryos, implying that the majority of expression changes were a direct consequence of altered Ubx chromatin interactions. A detailed analysis of biological GO terms showed that down-regulated genes mostly functioned in general cell processes like translation, RNA processing and gene expression (*Figure 3L*), while up-regulated genes controlled primarily lineage-specific functions like central nervous system, axon and cutical development (*Figure 3L*). To have a global view of the biological processes directly controlled by Ubx, we compared overrepresentation of higher-order biological GO terms between the Ubx degradation and control transcriptomes sets using the WEADE tool. We observed that processes were not randomly changed in the absence of Ubx but were mostly in agreement with Ubx chromatin binding (*Figure 3M*). For example, translation and cell cycle processes, which were enriched among the expressed as well as Ubx bound active genes in the control transcriptome, were now represented among the transcripts with reduced accumulation in the absence of Ubx (*Figure 3M*), supporting that these processes were directly activated by Ubx. On the other hand, stimulus and signalling related processes, which were only to a minor extent represented among the genes expressed in the mesoderm, were found enriched among the genes with enhanced activity in the absence of Ubx (*Figure 3M*). Consistent with a repressive function of Ubx, these processes were overrepresented among the inactive genes bound by Ubx in control mesodermal nuclei (*Figure 3M*).

Intriguingly, we found differentiation processes to be enriched among the genes up-regulated in the absence of Ubx (*Figure 3L,M*). Analysing differentiation processes revealed that processes controlling the development of lineages other than the mesoderm, including the neuronal, ectodermal

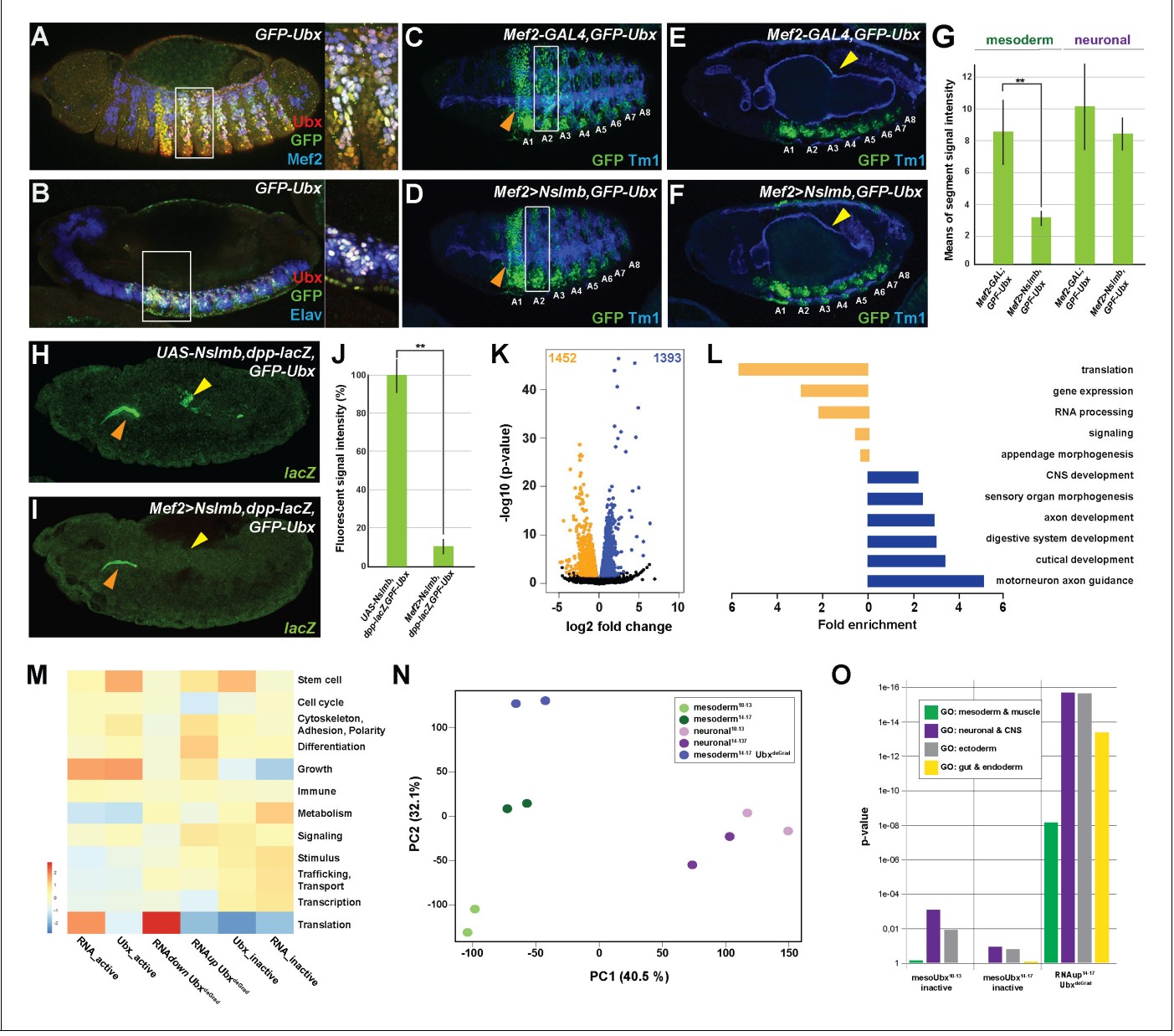

**Figure 3.** Ubx directly represses alternative fate genes in a lineage-specific manner. (A, B) Lateral view of stage 14 *GFP-Ubx Drosophila* embryos stained for the muscle differentiation marker Mef2 (blue) (A), the pan-neuronal marker Elav (blue) (B), Ubx (red) and GFP (green). Boxes indicate the location of the close-ups on the right panel. (C–F) Lateral view of stage 15 *Mef2-GAL4,GFP-Ubx* and *Mef2 >Nslmb,GFP-Ubx Drosophila* embryos stained for Tm1 (blue) to indicate the differentiated muscles and GFP (green) to highlight GFP-Ubx expression. The boxes in (C, D) mark GFP expression in muscle cells of the 2nd abdominal segment (A2), which is lost when Ubx is degraded, while the ectodermal expression in the 1st abdominal segment (A1) is unaffected (marked by orange arrowheads). In (E, F), GFP expression in the CNS and the visceral mesoderm (marked by yellow arrowheads) is shown. GFP expression in the visceral mesoderm is lost in *Mef2 >Nslmb,GFP-Ubx Drosophila* embryos (F), leading to a loss of the second midgut constriction. (G) Quantification of GFP signal intensity in the mesoderm and the CNS of *Mef2-GAL4,GFP-Ubx* and *Mef2 >Nslmb,GFP-Ubx Drosophila* embryos, showing that GFP is strongly decreased in the mesoderm (\*\*=p < 0.01). (H, I) *lacZ* mRNA expression in stage 14 *UAS-Nslmb, dpp-lacZ,GFP-Ubx* and *Mef2 >Nslmb,dpp-lacZ,GFP-Ubx Drosophila* embryos, highlighting that mesoderm-specific depletion of Ubx leads to a loss of *dpp-lacZ* enhancer activity (indicated by yellow arrowheads), a known and direct target of Ubx control (*Capovilla and Botas, 1998*; *Manak et al., 1994*). The orange arrowheads highlight unspecific enhancer activity in the salivary glands. (J) Quantification of *lacZ* signal intensity in *UAS-Nslmb,dpp-lacZ,GFP-Ubx* and *Mef2 >Nslmb,dpp-lacZ,GFP-Ubx* embryos, showing that *lacZ* expression is strongly decreased (\*\*=p < 0.01). (K) Volcano plot displaying the differentially expressed genes between *UAS-Nslmb,GFP-Ubx* and *Mef2 >Nslmb,GFP-Ubx* INTACT-sorted mesodermal nuclei. The y-axis corresponds to the mean expression value of log10 (p-value), and the x-axis displays the log2 fold change value. The orange dots represent transcripts whose expression is down-regulated (padj-value <0.1), the blue dots represent the up-regulated expressed transcripts (padj-value <0.1) between *UAS-Nslmb,GFP-Ubx* and *Mef2 >Nslmb,GFP-Ubx* INTACT-sorted mesodermal nuclei. (L) Fold enrichment of gene ontology terms of down- (orange) and up-

*Figure 3 continued on next page*

*Figure 3 continued*

regulated (blue) genes in *Mef2 >Nslmb,GFP-Ubx* vs. *UAS-Nslmb,GFP-Ubx* mesodermal nuclei. (**M**) Heat-map displaying presence of genes belonging to higher-order categories in the different gene classes. The colour range corresponds to the centred and scaled (per column) fraction of genes annotated to the category that also appear in the sample: red colour represents high values, blue colours low fractions of genes in the category, which are also present in the sample. Rows and columns are hierarchically clustered using Euclidean distance with complete linkage. (**N**) PCA applied to all RNA-seq samples identifies the separation of the Ubx$^{deGrad}$ dataset from the two mesodermal as well as the neuronal datasets. (**O**) Multiple testing of higher-order GO-terms related to different lineages among different gene classes. The y-axis corresponds to the p-value, the x-axis displays the different categories of tested genes. Neuronal and ectodermal as well as gut/endoderm related GO-terms are significantly enriched in the tested samples.

DOI: https://doi.org/10.7554/eLife.42675.007

The following figure supplement is available for figure 3:

**Figure supplement 1.** Generation and validation of CRIPR/Cas9 generated *GFP-Ubx* flies.

DOI: https://doi.org/10.7554/eLife.42675.008

end endodermal lineages, were predominantly represented among the up-regulated genes in Ubx depleted mesodermal cells (*Figure 3L*, *Figure 3—figure supplement 1P*), which was significantly higher than expected by chance (neuronal and ectoderm: p-value<2.2e-16, endoderm and gut structures: p-value 5.04e-14). Comparing these genes to the neuronal transcriptomes, which we have generated, showed that 56% (774/1393) of the up-regulated genes were normally expressed and functional in the neuronal lineage in the wild type situation (*Figure 3—figure supplement 1P*), while the remaining 44% (617/1393) were active in other lineages according to GO term prediction (data not shown).

Taken together, these results showed that Ubx controlled a large number of genes encoding diverse functions. The strong correlation between Ubx binding with specifically repressed or activated classes of genes suggested that Ubx indeed exerts a dominant influence over the mesodermal transcriptome at the two stages analysed. In addition, this analysis highlighted that Ubx repressed many genes controlling the establishment of non-mesodermal, alternative lineages, indicating that Ubx could have a pivotal role in restricting the developmental potential of cells and tissues, thereby ensuring that lineages adopt a unique identity.

## Ubx represses alternative fate genes by organizing the epigenetic landscape

One intriguing of this study finding was the high number of genes repressed by Ubx, which encoded primarily functions related to other lineages (*Figure 3L*). As this suggested that Ubx carves the developmental path of a cell or tissue lineage by inhibiting other possible developmental routes, we focused our subsequent analysis on the question how Ubx executes this lineage-restricting function. As gene expression critically depends on the epigenetic status of the gene regulatory control regions (*Kingston and Tamkun, 2014*; *Li et al., 2007*), we characterized Ubx chromatin interactions in more detail. To this end, we mapped two key modifications of the nucleosome component Histone H3, H3K27me3, a mark associated with repressive chromatin (*Filion et al., 2010*; *Kharchenko et al., 2011*), as well as H3K27ac, a mark associated with active promoters and enhancers (*Bonn et al., 2012*; *Rada-Iglesias et al., 2011*), by ChIPseq using INTACT-sorted mesodermal nuclei. The ChIP-seq for histone marks was performed in biological duplicates. This analysis revealed that in the early mesoderm about 30% of all Ubx chromatin interactions overlapped with H3K27me3 peaks and about 20% with H3K27ac peaks, while in the late mesoderm 40% of all Ubx chromatin interactions overlapped with H3K27me3 peaks and 73% with H3K27ac peaks (*Figure 4— figure supplement 1B*). Ubx interactions with repressive chromatin were predictive of gene activity, as about 70% of these occurred at inactive genes irrespective of the developmental stage (*Figure 4B*), which we found again associated with functions unrelated to the mesoderm (neuronal: p-value 7.05e-5–6.59e-6, ectoderm: p-value 4.80e-5–5.21e-7) (*Figure 4C*). In contrast, less than half of the Ubx interactions with active chromatin occurred close to active genes (*Figure 4B*), which were as before enriched for mesoderm-related functions (*Figure 4C*), while the rest were found close to inactive genes (*Figure 4B*). Organizing chromatin environment by genomic location revealed that only a minor fraction (about 20%) of the Ubx bound repressive chromatin regions occurred at

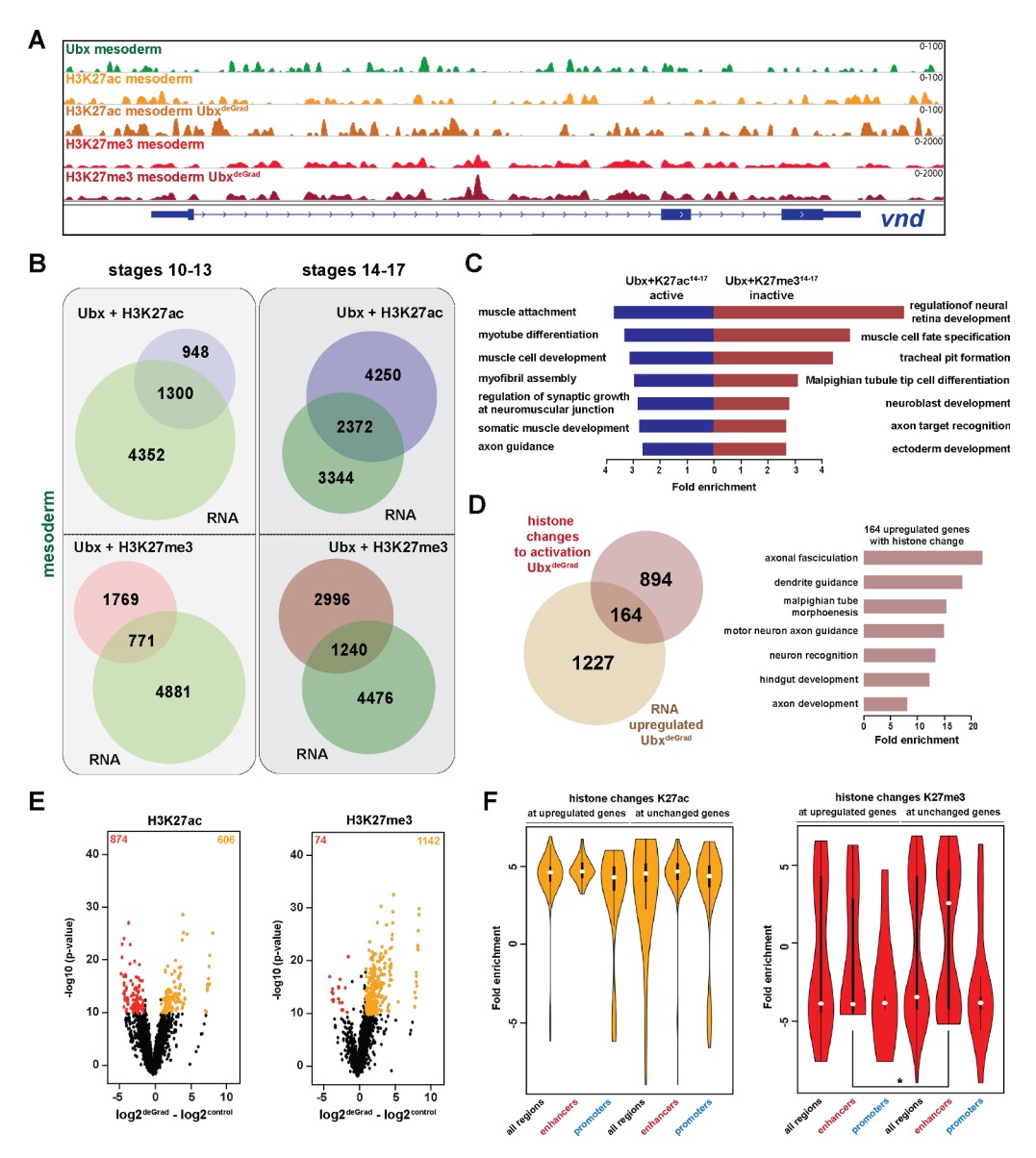

**Figure 4.** Ubx represses alternative fate genes by organizing the epigenetic landscape. (A) ChIP-seq binding profiles of Ubx (green), H3K27ac (light +dark orange) and H3K27me3 (light +dark red) in control (*UAS-Nslmb,GFP-Ubx*) (light orange +red) and Ubx^DeGrad (*Mef2 >Nslmb,GFP-Ubx*) (dark orange +red) mesodermal nuclei at the *vnd* genomic locus (blue), a gene up-regulated in the absence of Ubx. (B) Venn diagrams representing the overlaps between genes expressed in the mesoderm (green), which are targeted by Ubx and simultaneously associated with H3K27ac (Ubx +H3K27ac, purple) or H3K27me3 (Ubx +H3K27me3, red) marks, respectively. (C) Fold enrichment of gene ontology terms among the genes expressed, targeted by Ubx and associated with H3K27ac marks (Ubx +K27ac^14-17 active, blue) and the genes not expressed in the mesoderm, targeted by Ubx and associated with H3K27me3 marks (Ubx +K27me3^14-17 inactive, red). (D) Left panel: Venn diagram representing the overlap between genes up-regulated in Ubx depleted mesodermal nuclei (beige) and genes associated with histone mark changes indicative for gene activation (gain/increase in H3K27 acetylation, loss/reduction in H3K27 tri-methylation) (brown). Right panel: Fold enrichment of gene ontology terms of the 164 genes, which have histone changes to activation and are up-regulated in Ubx depleted mesodermal nuclei. (E) Volcano plot displaying the changes of H3K27ac and H3K27me3 marks between *UAS-Nslmb,GFP-Ubx* (control) and *Mef2 >Nslmb,GFP-Ubx* (degrade) INTACT-sorted mesodermal nuclei using differential binding analysis of ChIP-Seq peaks (DiffBind). The y-axis corresponds to the mean expression value of log10 (p-value), and the x-axis displays the difference of the log2 fold change value in Ubx^Degrad compared to the log2 of the control. The orange dots represent genomic regions containing an increase in H3K27ac or H3K27me3 (p-value<0.05), the red dots represent genomic regions containing a decrease in H3K27ac or H3K27me3 (p-value<0.05) between *UAS-Nslmb,GFP-Ubx* and *Mef2 >Nslmb,GFP-Ubx* INTACT-sorted mesodermal nuclei. (F) Comparison of H3K27ac or H3K27me3 fold enrichments at Ubx targeted genomic regions that experienced histone changes towards activation represented as Violin plots. Quantification of the histone changes in Ubx bound upregulated (164) and unchanged genes (894) by using the results of the differential binding analysis of ChIP-Seq peaks. The y-axis displays

*Figure 4 continued on next page*

*Figure 4 continued*

the fold change calculated by DiffBind. The x-axis shows all selected regions (upregulated and unchanged) and a sub-division of the regions in promoter (−1000 to +10 from TSS, 5' UTR) and enhancer (distal enhancers (−2000 to −1000 from TSS), 3' UTR, downstream, intronic regions, distal intergenic). Each sample contains a significant increase in H3K27ac levels (yellow), H3K27me3 levels (red) are decreased, but enhancer regions close to genes not changed in their expression have significantly higher H3K27me3 levels than enhancers close to genes up-regulated after Ubx depletion (*=p < 0.05).

DOI: https://doi.org/10.7554/eLife.42675.009

The following figure supplement is available for figure 4:

**Figure supplement 1.** Ubx represses alternative fate genes by organizing the epigenetic landscape.

DOI: https://doi.org/10.7554/eLife.42675.010

promoters, while the majority was found at putative distal enhancers, in introns and intergenic regions in both developmental time points (*Figure 4—figure supplement 1D*).

In a next step, we asked how the epigenetic landscape changed in the absence of Ubx, in particular at genomic regions with overlapping Ubx and repressive H3K27me3 marks. To this end, we performed ChIP-seq experiments in biological triplicates, the two most similar duplicates were used for further analysis. In total, 1216 H3K27me3 peaks and 1480 H3K27ac peaks located in gene regulatory regions, which corresponded to 2768 genes, changed their chromatin state in the absence of Ubx, as they experienced either a reduction/loss or an increase/gain in tri-methylation/acetylation at lysine 27 of histone 3 (*Figure 4E*, *Figure 4—figure supplement 1E*). Importantly, 80% of these genomic regions possessed an Ubx binding peak in control mesodermal cells, indicating that Ubx was critically required for establishing or maintaining the majority of these histone marks. Intriguingly, 1058 of these genes were primed for gene expression, as they experienced a loss/reduction of H3K27me3 and/or gain/increase of H3K27ac in their regulatory regions (*Figure 4D*), and 164 of them were indeed up-regulated in the mesodermal lineage in the absence of Ubx (*Figure 4D*, *Figure 4—figure supplement 1C*). Consistent with previous results, these 164 genes were highly enriched for processes critically controlling the development of lineages other than the mesoderm (neuronal: p-value<2.2e-16, ectoderm: p-value 1.36e-12, endoderm and gut structures: p-value<2.2e-16) (*Figure 4D*).

One result that puzzled us was that the majority of genes primed for activation (894 of 1058) remained silent on the RNA level when Ubx was lineage-specifically degraded (*Figure 4D*). In order to resolve this discrepancy, we analysed H3K27me3 and H3K27ac changes at introns, intergenic and distal enhancer regions, which we collectively labelled enhancers (*Figure 4F*), as well as at promoters. We found that H3K27ac marks were equally increased at promoters and enhancers of genes unchanged or up-regulated in the absence of Ubx (*Figure 4F*). However, while H3K27me3 marks were significantly decreased at enhancers as well as promoters of up-regulated genes, unchanged genes showed a significant increase in H3K27me3 at their enhancers (*Figure 4F*). As a high proportion of Ubx and H3K27me3 peaks co-occurred at enhancers (*Figure 4—figure supplement 1D*), this result revealed that Ubx played a major role in repressing the expression of alternative fate genes by controlling the deposition/maintenance of H3K27me3 marks at enhancers.

In sum, these results showed that Ubx mediated the repression of genes encoding non-mesodermal functions by controlling the epigenetic status, in particular H3K27me3, at Ubx bound chromatin sites located in enhancers.

## Ubx interacts with Pho at H3K27me3 marked inactive genes encoding alternative fate genes

To identify factors that together with Ubx could mediate the repression of alternative fate genes thereby restricting lineage identity, we performed a DNA motif search using all Ubx peaks either overlapping with H3K27me3 repressive or H3K27ac active marks. We found in both cases motifs for Ubx, Extradenticle (Exd), a TALE class homeobox TF functioning as Hox cofactor in invertebrates and vertebrates (*Mann et al., 2009*; *Moens and Selleri, 2006*), and Trithorax-like (Trl), a GAGA factor activating and repressing gene expression by chromatin modification (*Kingston and Tamkun, 2014*), among the highest ranking motifs. In contrast, the DNA binding motif for the zinc finger protein Pleiohomeotic (Pho) was specifically enriched only among the co-occurring Ubx and H3K27me3 binding events (*Figure 5A*). Interestingly, it has been shown just recently that in the absence of Pho,

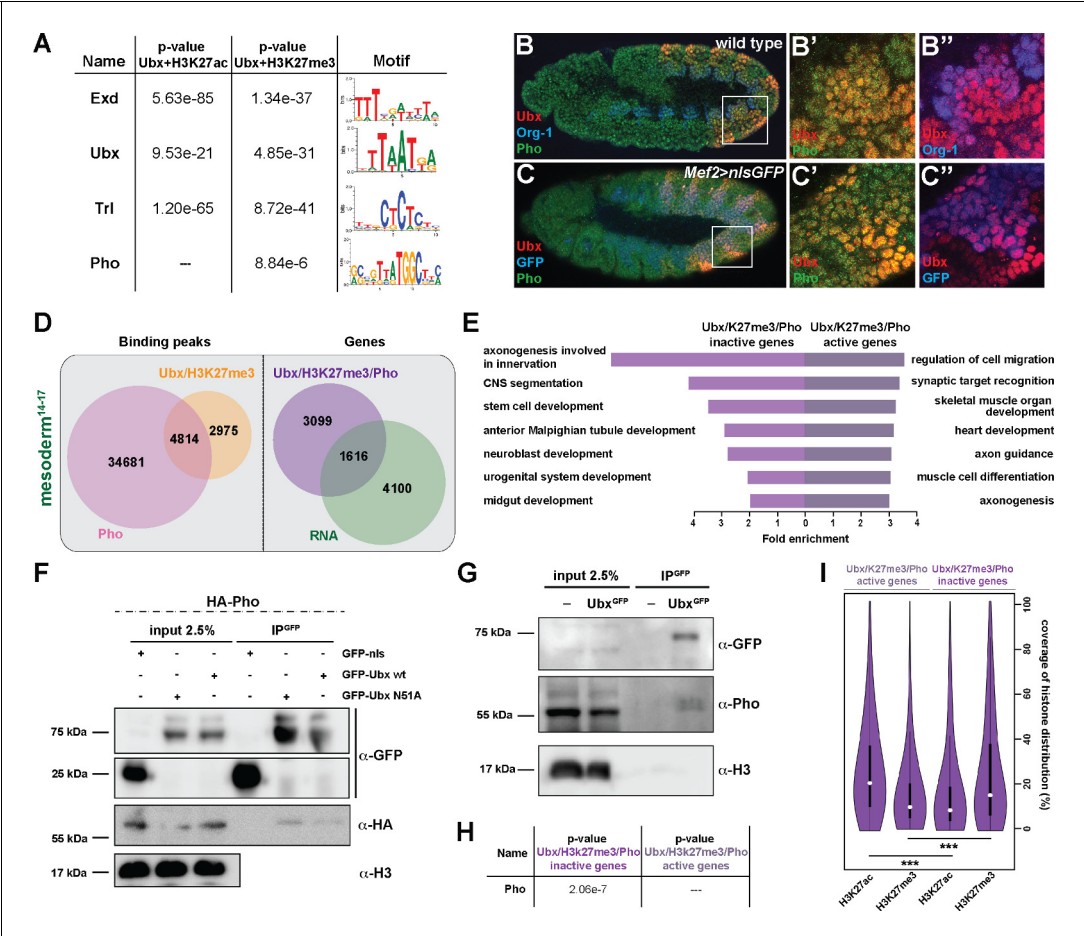

**Figure 5.** Ubx interacts with Pho at H3K27me3 marked inactive genes encoding alternative fate genes. (**A**) The top four hits from MEME-ChIP analysis using all Ubx targeted, H3K27ac marked genomic regions in the vicinity of expressed genes (Ubx +H3K27ac) or Ubx targeted, H3K27me3 marked genomic regions in the vicinity of inactive genes (Ubx +H3K27me3) as input. While the Ubx, Exd and Trl motifs were found over-represented in both datasets, the Pho motif was found over-represented only in the Ubx +H3K27me3 dataset. (**B–B''**) Lateral view of stage 11 wild-type *Drosophila* embryos stained for the mesodermal marker Org-1 (blue), Ubx (red) and Pho (green). (**B'**) and (**B''**) are dual colour images of (**B**), highlighting that Ubx and Pho are co-expressed in Org-1 labelled mesodermal cells. (**C–C''**) Lateral view of stage 11 *Mef2 >nlsGFP Drosophila* embryos stained for GFP (blue), Ubx (red) and Pho (green). (**C'**) and (**C''**) are dual colour images of (**C**). (**D**) Left: Venn diagram representing the overlap between Ubx targeted and H3K27me3 marked chromatin regions (orange) and Pho binding peaks (pink) in stage 14–17 mesodermal nuclei. Right: Venn diagram representing the overlap between genes targeted by Ubx +Pho and marked by H3K27me3 (purple) and the genes expressed in the mesoderm (green). (**E**) Fold enrichment of gene ontology terms of the genes targeted by Ubx and Pho, which are additionally marked by H3K27me3 and either not expressed (3578) or expressed (1137) in mesodermal cells. (**F**) Co-immunoprecipitation of exogenous GFP fusion proteins using GFP-trap beads after transfection of S2R + *Drosophila* cells with HA-Pho coupled with GFP-nls (negative control), GFP-Ubx WT or N51A (mutant of the homeodomain, Asparagine 51 replaced by Alanine residue). Western blots were probed with the indicated antibodies. Pho is detected in the immunoprecipitated fraction of GFP-Ubx WT and N51A, while it is absent in the GFP negative control. (**G**) Co-immunoprecipitation with the GFP-trap system of endogenous proteins performed on nuclear extract from embryos expressing (or not) the endogenous GFP-Ubx fusion protein. Pho co-immunoprecipitates with GFP-Ubx, while it is absent from w1118 fly line. (**H**) MEME-ChIP analysis using Ubx and Pho targeted chromatin regions marked by H3K27me3 located in the vicinity of active or inactive genes identifies the classical Pho motif over-represented only in the vicinity of inactive genes. (**I**) Quantification of H3K27ac and H3K27me3 distributions at Ubx and Pho targeted chromatin regions marked by H3K27me3 located either in the vicinity of active or inactive genes are represented as Violin plot. The y-axis displays the coverage of the histone mark distribution in percent, the x-axis indicates the histone mark that was analysed. Active genes display a higher coverage of H3K27ac, while inactive genes have a higher coverage of H3K27me3 (***=p < 0.001).

DOI: https://doi.org/10.7554/eLife.42675.011

which recruits PcGs to PREs (*Frey et al., 2016*; *Fritsch et al., 1999*; *Kahn et al., 2014*), H3K27me3 marks were reduced in Polycomb regions and redistributed to heterochromatin (*Brown et al., 2018*). In combination with our finding that H3K27me3 levels were reduced specifically at up-regulated genes (*Figure 4F*), we hypothesized that Pho could function together with Ubx in the lineage-

specific repression of alternative fate genes. Consistent with this idea Pho was found expressed in mesodermal cells also expressing Ubx (*Figure 5B-C''*). Furthermore, we confirmed a weak interaction of Ubx and Pho proteins in a complex in vitro and in vivo by performing co-immunoprecipitation (Co-IP) experiments *in cellulo* using *Drosophila* S2R + cells transfected with tagged versions of Ubx and Pho as well as in vivo using GFP-Ubx embryos (*Figure 5F,G*). This result suggested that Ubx and Pho could interact with the same chromatin regions to control gene expression, thus we analysed high-resolution Pho maps retrieved from embryonic (stage 9 to 12) mesodermal cells (*Erceg et al., 2017*). We found 4814 chromatin regions (which represent 62% of the Ubx and 12% of the Pho binding events) to be co-bound by Ubx and Pho and marked by H3K27me3 in stage 14–17 mesodermal cells (*Figures 5D* and *6A*), with 40% of these events occurring at promoters and 49% at enhancers (data not shown). Consistent with the reported role of Pho in transcriptional repression (*Frey et al., 2016*; *Fritsch et al., 1999*; *Kahn et al., 2014*), 66% (3099/4715) of the genes bound by Ubx and Pho as well as marked by H3K27me3 were not expressed in the mesoderm (*Figure 5D*), and GO term analysis revealed that the majority of the genes encoded non-mesodermal and stem cell-related functions (neuronal: p-value 1,62e-4, ectodermal: 1,71e-5) (*Figure 5E*). This was different for the remaining 1616 genes also bound by Ubx and Pho and marked by H3K27me3 but expressed in mesodermal cells, as they encoded gene functions controlling processes typical for the tissue type and developmental stage (*Figure 5E*). By analysing the distribution of H3K27me3 and H3K27ac at regulatory regions, we discovered that those chromatin regions associated with inactive genes had a higher coverage of H3K27me3 and lower coverage of H3K27ac at shared Ubx/Pho binding regions, while it was the opposite for active genes (*Figure 5I*). Furthermore, the canonical Pho binding motif was found overrepresented only among the Ubx/Pho/H3K27me3 chromatin regions associated with genes inactive in the mesoderm (*Figure 5H*), while other DNA binding motifs, including the ones for Ubx, Exd and Trl, were overrepresented in Ubx/Pho/H3K27me3 chromatin regions associated with inactive as well as active genes.

These results demonstrated that Ubx interacted with Pho and that their interaction on H3K27me3 marked chromatin regions occurred preferentially at inactive genes in mesodermal cells.

## Ubx is required for stabilizing pho binding to H3K27me3 chromatin regions

Our results indicated that Ubx could mediate the repression of alternative fate genes by recruiting or stabilizing Pho binding at Ubx targeted genomic regions in the mesodermal lineage, thereby allowing the two proteins to act in a combinatorial fashion. In support of this hypothesis, we found 55% (769/1391) of the genes up-regulated in the absence of Ubx, which were highly enriched for non-mesodermal and stem cell related functions, to be located in the vicinity of Ubx/Pho binding regions (*Figure 6—figure supplement 1A*). In contrast, only a minor fraction (15%) was in the vicinity of chromatin sites targeted by Pho only, and we did not detect any significant enrichment of GO terms among these genes (data not shown). To provide additional evidence for a combined action of Ubx and Pho, we analysed the binding of Pho to Ubx/Pho/H3K27me3 binding regions in the absence of Ubx by performing ChIP experiments on INTACT sorted control and Ubx depleted mesodermal nuclei. Furthermore, we also quantified the levels of H3K27me3 and H3K27ac marks at these loci. We selected five Ubx/Pho/H3K27me3 genomic regions that changed their histone marks towards activation (less H3K27me3 and/or more H3K27ac) when Ubx was depleted in the mesodermal lineage, which resulted in the activation of the associated genes encoding neuronal functions. This included *ventral nervous defective* (*vnd*), a NK2 class TF encoding gene critical for patterning of the neuroectoderm as well as the formation and specification of ventral neuroblasts (*Skeath et al., 1994*), *HGTX*, a homeodomain TF encoding gene promoting the specification and differentiation of motor neurons innervating the ventral body wall muscles (*Syu et al., 2009*), *Neurotrophin 1* (*NT1*), a cytokine encoding a gene that regulates motor neuron survival and axon guidance, *hamlet* (*ham*), a gene encoding a PRDM class TF that regulates neuron fate selection in the peripheral nervous system (*Moore et al., 2002*), and *Ptx1*, again a homeodomain TF encoding gene which is expressed at high levels in the early embryonic central nervous system and the midgut, and later also in ventral muscles (*Vorbrüggen et al., 1997*). In addition, we also chose five loci bound by Pho but not by Ubx, which were associated with genes normally not expressed in the mesoderm and which remained silent in the absence of Ubx, including *folded gastrulation* (*fog*), *Olig family* (*Oli*), *bendless* (*ben*), *myospheroid* (*mys*) and *medial glomeruli* (*meigo*). Strikingly, all five Ubx/Pho/

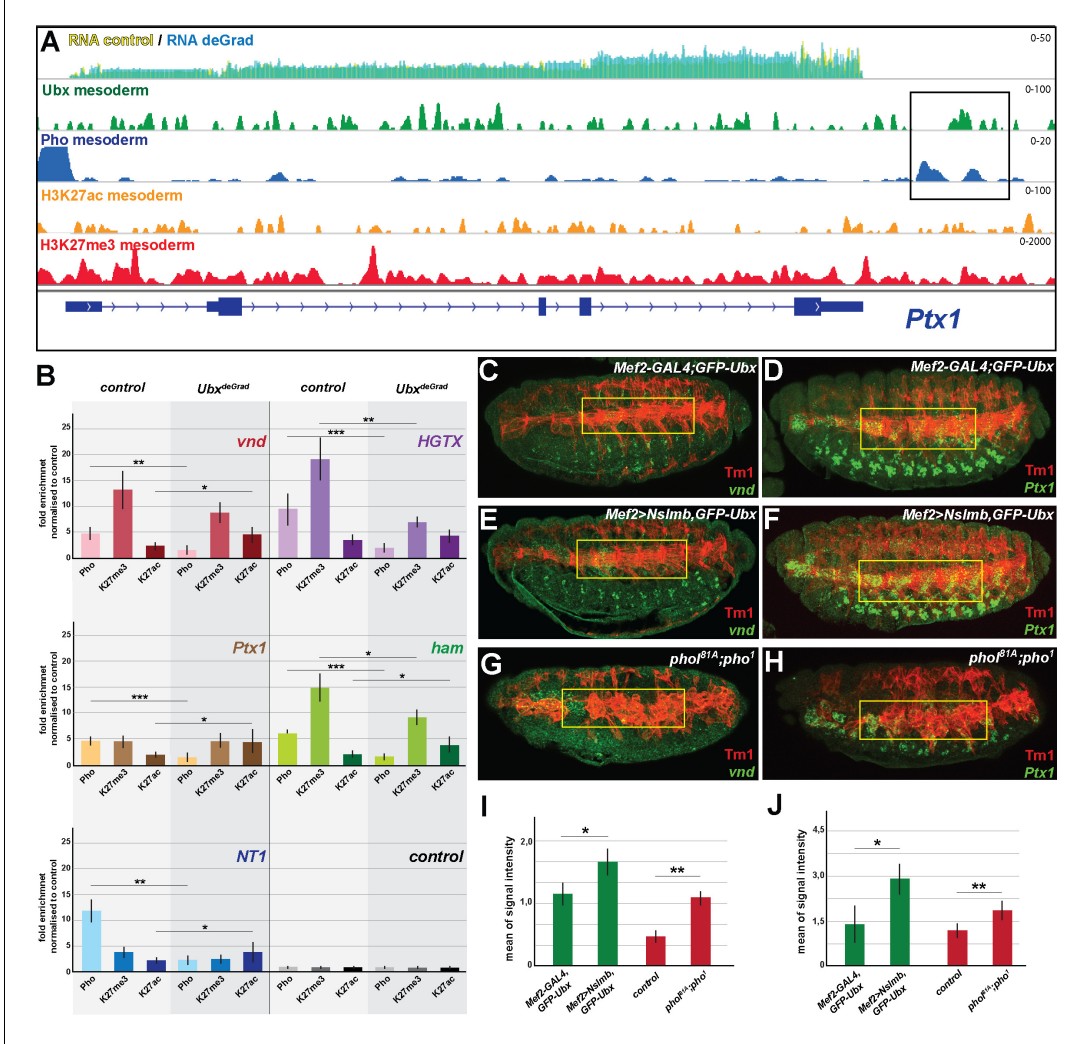

**Figure 6.** Ubx is required for stabilizing Pho binding to H3K27me3 chromatin regions. (A) ChIP-seq binding profiles of Ubx (green), Pho (Pho), H3K27ac (orange) and H3K27me3 (red) in mesodermal nuclei as well as RNA-seq profiles in *Mef2-GAL4,GFP-Ubx* control (yellow) versus and *Mef2 >Nslmb,GFP-Ubx* (blue) mesodermal nuclei at the *Ptx1* genomic locus. The *Ptx1* gene are shown in blue. The box highlights the overlapping binding peaks of Ubx and Pho in the vicinity of the *Ptx1* coding region. (B) ChIP-qPCR experiments for Pho, H3K27ac and H3K27me3 using chromatin regions close to the *vnd, HGTX, Ptx1, ham* and *NT1* genes bound by Pho and Ubx using chromatin isolated from *UAS-Nslmb,GFP-Ubx* (light green) and *Mef2 >Nslmb,GFP-Ubx* (dark green) mesodermal nuclei. As control locus sites, a exon region in the *scramb* genes was used, which is not bound by Pho nor by Ubx. All loci show a significant reduction of Pho binding, H3K27me3 levels were reduced at the HGTX associated Ubx/Pho chromatin regions. In the *vnd, Ptx1, ham* and NT1 associated Ubx/Pho loci H3K27ac levels are significantly enriched (*=p < 0.05, **=p < 0.01, ***=p < 0.001). (C–H) Lateral view of stage 16 *Mef2-GAL4,GFP-Ubx* (C, D), *Mef2 >Nslmb,GFP-Ubx* (E, F) and *phol^{81A};pho^1* mutant (G, H) embryos stained for the muscle marker Tm1 (C–H) and for *vnd* (C, E, G) and *Ptx1* (D, F, H) transcripts. The yellow boxes highlight the lateral muscles. (I, J) Quantification of *vnd* (I) and *Ptx1* (J) signal intensities in the lateral muscles (as indicated by the yellow boxes) in *Mef2-GAL4,GFP-Ubx, Mef2 >Nslmb,GFP-Ubx* and *phol^{81A};pho^1* mutant embryos, showing that the expression of *vnd* and *Ptx1* is significantly increased in comparison to control embryos (*=p < 0.05).

DOI: https://doi.org/10.7554/eLife.42675.012

The following figure supplement is available for figure 6:

**Figure supplement 1.** Ubx interacts with Pho at inactive genes encoding alternative fate genes.
DOI: https://doi.org/10.7554/eLife.42675.013

H3K27me3 loci, which were co-bound by Ubx and Pho in the presence of Ubx (*Figure 6A, Figure 6—figure supplement 1B*), experienced a dramatic loss of Pho binding when Ubx was degraded (*Figure 6B*). In contrast, Pho binding to Pho-only control loci remained unaffected (*Figure 6—figure supplement 1C*). As Pho expression levels were unaltered in the absence of Ubx (*Figure 7C*), we concluded that Ubx is required to stabilize Pho binding to chromatin of H3K27me3

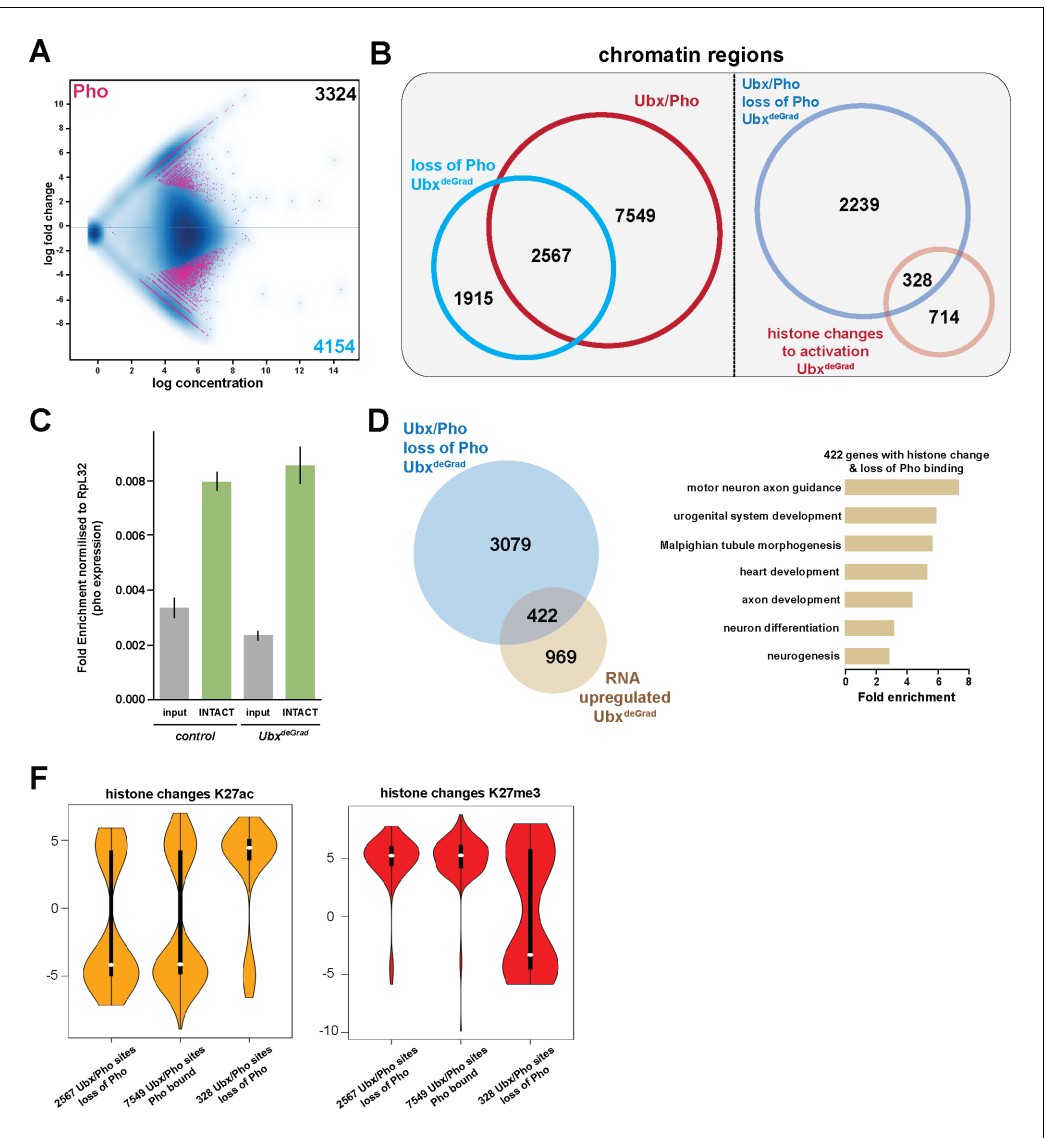

**Figure 7.** General relevance of Pho dependency on Ubx. (**A**) MA plot displaying the changes of Pho binding between *UAS-Nslmb,GFP-Ubx* (control) and *Mef2 >Nslmb;GFP-Ubx* (Ubx^deGrad) INTACT-sorted mesodermal nuclei using differential binding analysis of ChIP-Seq peaks. The y-axis corresponds to the log fold change, and the x-axis displays the log concentration. (**B**) Left panel: Venn diagram representing an overlap between genomic regions co-bound by Ubx and Pho (Ubx/Pho, red) and regions that lose Pho binding in the Ubx^deGrad background. Right panel: Venn diagram illustrating the overlap between genomic regions co-bound by Ubx and Pho that experience a loss of Pho binding in the Ubx^deGrad (light blue) and regions containing a histone mark change towards activation in the Ubx depleted mesodermal nuclei (light red). (**C**) Expression of *pho* in in *UAS-Nslmb;GFP-Ubx* (control) and *Mef2 >Nslmb;GFP-Ubx* (Ubx^deGrad) mesodermal nuclei, fold change is normalized to Ribosomal protein L32 (RpL32). The expression of *pho* is not significantly changed in the mesodermal, INTACT- sorted nuclei (INTACT) upon Ubx depletion (Ubx^deGrad) in comparison to the control sample. (**D**) Left panel: Venn diagram representing the overlap between genes up-regulated in Ubx depleted mesodermal nuclei (yellow) and genes associated with Ubx and Pho that experience a loss of Pho binding in the Ubx^deGrad (blue). Right panel: Fold enrichment of gene ontology terms of the 422 genes, which contain Ubx and Pho binding, lose Pho binding and are up-regulated in Ubx depleted mesodermal nuclei. (**F**) Comparison of H3K27ac or H3K27me3 fold enrichments at Ubx and Pho targeted genomic regions that experienced a loss of Pho binding, represented as Violin plots. Quantification of the different regions: 1) Ubx/Pho sites that lose Pho binding (2597), 2) Ubx/Pho sites that maintain Pho binding (7549), 3) Ubx/Pho sites that lose Pho binding and experiences a histone mark change (328). The y-axis displays the fold change calculated by DiffBind. The x-axis shows all selected regions as disrobed 1), 2)
*Figure 7 continued on next page*

*Figure 7 continued*

and 3). Regions that lose Pho binding and experiences a histone mark change (328) experience a significant increase in H3K27ac levels (yellow) and a decrease in H3K27me3 levels (red).

DOI: https://doi.org/10.7554/eLife.42675.014

marked loci. These results indicated that the expression of alternative fate genes should be similarly de-repressed in the mesoderm in the absence of either Ubx or Pho. In line with this hypothesis, we found the expression of *vnd* and *Ptx1*, which were normally not or only weakly expressed in mesodermal cells of stage 16 control embryos (*Figure 6C,D,I,J*) to be significantly increased in the mesoderm of *Mef2 >Nslmb,GFP-Ubx* embryos (*Figure 6E,F,I,J*, *Figure 6—figure supplement 1D,E*) as well as in $phol^{81A}/pho^1$ double mutant embryos (*Figure 6G,H,I,J*). It has been recently reported that in the absence of Pho H3K27me3 enrichment was decreased (*Brown et al., 2018*; *Kraushaar et al., 2013*), and consistently, we found that H3K27me3 levels at the *HGTX* and *ham* associated Ubx/Pho/H3K27me3 loci were significantly decreased in Ubx depleted mesodermal nuclei (*Figure 6B*). Importantly, while H3K27ac levels remained unaffected at the *HGTX* associated Ubx binding locus, they were increased at the *ham* associated Ubx/Pho/H3K27me3 region (*Figure 6B*). In contrast, H3K27me3 levels at Ubx/Pho/H3K27me3 loci associated with the *vnd*, *NT1* and *Ptx1* genes were not considerably decreased, however, these loci had significantly higher H3K27ac levels in the absence of Ubx (*Figure 6B*). These results revealed that in the absence of Ubx Pho's ability to interact with the five Ubx/Pho/H3K27me3 loci was strongly reduced and concomitantly H3K27me3 as well as H3K27ac levels were changed.

## General relevance of Pho dependency on Ubx

In a next step, we asked whether stabilization of Pho binding and maintenance of H3K27me3 and H3K27ac levels are generally dependent on Ubx. To this end, we performed ChIP-seq experiments in biological triplicates using a Pho antibody (*Klymenko et al., 2006*) and INTACT-sorted mesodermal nuclei that were depleted for Ubx protein. In total, 4154 sites experienced a loss/reduction in Pho binding in the absence of Ubx (*Figure 7A*), while at 3324 sites Pho binding was gained or increased (*Figure 7A*). Intriguingly, 62% (2567/4154) of the regions, which lost Pho binding in the absence of Ubx, were co-bound by Ubx in control nuclei (*Figure 7B*), revealing a co-dependency of Ubx and Pho binding to shared chromatin sites. Notably, 422 of the 3501 genes associated with Ubx/Pho regions that lost Pho binding were up-regulated in Ubx depleted mesodermal nuclei (*Figure 7D*), which again encoded preferentially proteins controlling alternative fates (*Figure 7D*). Our analysis of single Ubx/Pho co-bound loci revealed that the inability of Pho to remain bound at Ubx/Pho sites in the absence of Ubx resulted in a significant change of H3K27me3 as well as H3K27ac levels at these regions. Thus, we asked whether H3K27me3 and H3K27ac levels were generally altered at Ubx/Pho binding sites when Ubx was depleted in the mesodermal lineage. To this end, we determined H3K27ac and H3K27me3 levels at Ubx/Pho co-bound sites that lost Pho binding when Ubx was degraded. Notably, we did not detect any significant change of H3K27ac and H3K27me3 levels at these sites on the global level, which was also the case for Ubx/Pho co-bound sites that did not experience a change in Pho binding when Ubx was degraded (*Figure 7F*). However, a subset of 328 loci, which lost Pho binding and were defined before as regions experiencing histone changes towards activation (*Figure 4D*), showed a highly significant loss of H3K27me3 marks (as well as a gain of H3K27ac marks) (*Figure 7F*), demonstrating that a substantial number of Ubx/ Pho sites required Ubx for stabilizing Pho binding and H3K27me3 marks.

In sum, these results demonstrated that Ubx stabilized Pho binding to chromatin regions at alternative fate genes in the mesodermal lineage, which controlled the proper levels of H3K27me3 and H3K27ac marks, thereby ensuring repression of these genes.

## Discussion

Cell and tissue types get different during development and their identity needs to be maintained also in adulthood to guarantee the survival of organisms. Here, we provide evidence that multi-lineage TFs of the Hox class stabilize the different lineage choices by restricting cellular plasticity in a

lineage specific manner. To this end, we studied the broadly expressed Hox TF Ubx in the meso-dermal and neuronal lineages during *Drosophila* development using a comparative genomic approach and an experimental system to deplete Ubx protein exclusively in the embryonic meso-dermal lineage. This approach allowed us to dissect the cell-autonomous function of Ubx in a single lineage that was located in an otherwise normal cellular environment at the transcriptome and chro-matin level. Using this experimental set-up, we found that Ubx comprehensively orchestrates the transcriptional programs of the mesodermal as well as of the neuronal lineage, as it bound and regu-lated a substantially fraction of genes specifically expressed in these tissue lineages. Strikingly, this analysis revealed that the majority of Ubx chromatin interactions were located in the vicinity of inac-tive genes, and lineage-specific interference with Ubx in the mesoderm demonstrated that about 20% of these interactions were important for repressing the close-by genes. Intriguingly, these genes were highly enriched for alternative cell fates, demonstrating that Ubx had indeed a pivotal role in restricting the developmental potential of cell and tissue types in a context-dependent manner.

One important question arising from this result was why not more of the inactive genes bound by Ubx were de-repressed in the absence of Ubx. There are several explanations for this finding. First, Hox TFs cross-regulate each other's expression, a phenomenon described as posterior suppression (*Morata and Kerridge, 1982*; *Struhl, 1982*). Consistently, the *Hox* gene *Antennapedia* (*Antp*), which is normally expressed anterior to Ubx, is ectopically activated when Ubx function is absent (*Fig-ure 8—figure supplement 1*), allowing Antp now to partially take over the function of Ubx in this lineage. Second, gene regulation is tightly linked to the chromatin status at promoter and enhancer regions. It had been shown in mammalian cells that the turn-over rates of the histone variant H3.3 at regulatory regions were correlated with specific histone modifications, high turn-over when associ-ated with high levels of active histone modifications, like H3K27ac, while much slower turn-over when associated with higher levels of H3K27me3 marks (*Kraushaar et al., 2013*). This implies that the 'clearing' of repressive histone modifications takes much longer in comparison to active ones. Third, H3K27me3 marks serve as epigenetic memory to permanently silence genes in the course of development, and a recent study demonstrated that a resetting of the epigenetic status requires cell division to dilute the H3K27me3 mediated silencing effect (*Coleman and Struhl, 2017*). In line with these studies, we found that the reduction of H3K27me3 levels was low in comparison to the increase of H3K27ac levels in Ubx depleted mesodermal nuclei. Thus, a de-repression of genes might require either more time or cell divisions or both. However, after its specification at embryonic stage 12 the mesoderm does not divide anymore, which might prevent an efficient clearing of H3K27me3 marks at Ubx targeted chromatin sites. Fourth, de-repression of genes is not only a con-sequence of abolishing repression but also of gaining activation, which not only requires a change of the histone environment at control regions but also the expression and action of the proper sets of TFs. Comparing the transcriptomes of the mesodermal and neuronal lineages revealed that only about 5% of the TFs expressed in an alternative lineage, in this case the neuronal one, were expressed in the Ubx depleted mesodermal cells, including vnd, HGTX, ham and Ptx1, which was obviously not sufficient to induce a lineage switch, in particular as the expression of only about 10% of the mesodermal fate related TFs was reduced in their expression. As we find Antp to be ectopi-cally expressed in the mesodermal lineage in the absence of Ubx (*Figure 8—figure supplement 1*), we assume Antp to partially take over the lineage-specific function of Ubx. Thus, it will be interesting in future to study mesoderm development in a Hox-free environment and determine the fate of the developing cell lineage.

Another highly relevant finding of our study is that the Hox TF Ubx lineage-specifically repressed the transcription of alternative fate genes by organizing the epigenetic landscape and that chromatin changes at Ubx sites were dependent on the interaction of Ubx with the Polycomb recruiter Pho (*Henry et al., 2012*; *Schoenfelder et al., 2018*). Strikingly, our study revealed that Ubx was crucial for stabilizing binding of the PcG protein Pho to specific chromatin regions. Although it is known that DNA binding TFs other than Pho interact with PREs, for example Grainyhead (*Kassis and Brown, 2013*; *Müller and Kassis, 2006*), a TF that has been shown recently to lineage-specifically displace nucleosomes at enhancers (*Jacobs et al., 2018*), these studies were mostly performed in vitro and the role of these TFs in Pho chromatin targeting was not addressed (*Nishioka et al., 2018*). In contrast, we analysed the combinatorial interaction of Ubx and Pho on the chromatin in vivo, in the mesodermal lineage of *Drosophila* embryos. Importantly, we showed that Pho was no

longer able to interact with Ubx bound sites when Ubx protein was depleted in a lineage-specific manner, while Pho-only chromatin interactions were unaffected. As a result, H3K27me3 levels were reduced and H3K27ac levels increased, resulting in the de-repression of genes primarily associated with Ubx-Pho chromatin regions. This result indicated that not only Pho but very likely the whole PcG protein complex was not properly targeted to Ubx chromatin sites, which is in line with our finding that H3K27me3 levels were generally decreased at enhancers of genes de-repressed upon Ubx depletion (*Figure 8*). However, we also found H3K27ac levels increased at Ubx chromatin interactions located in the vicinity of de-repressed genes. Interestingly, PREs frequently co-localize with

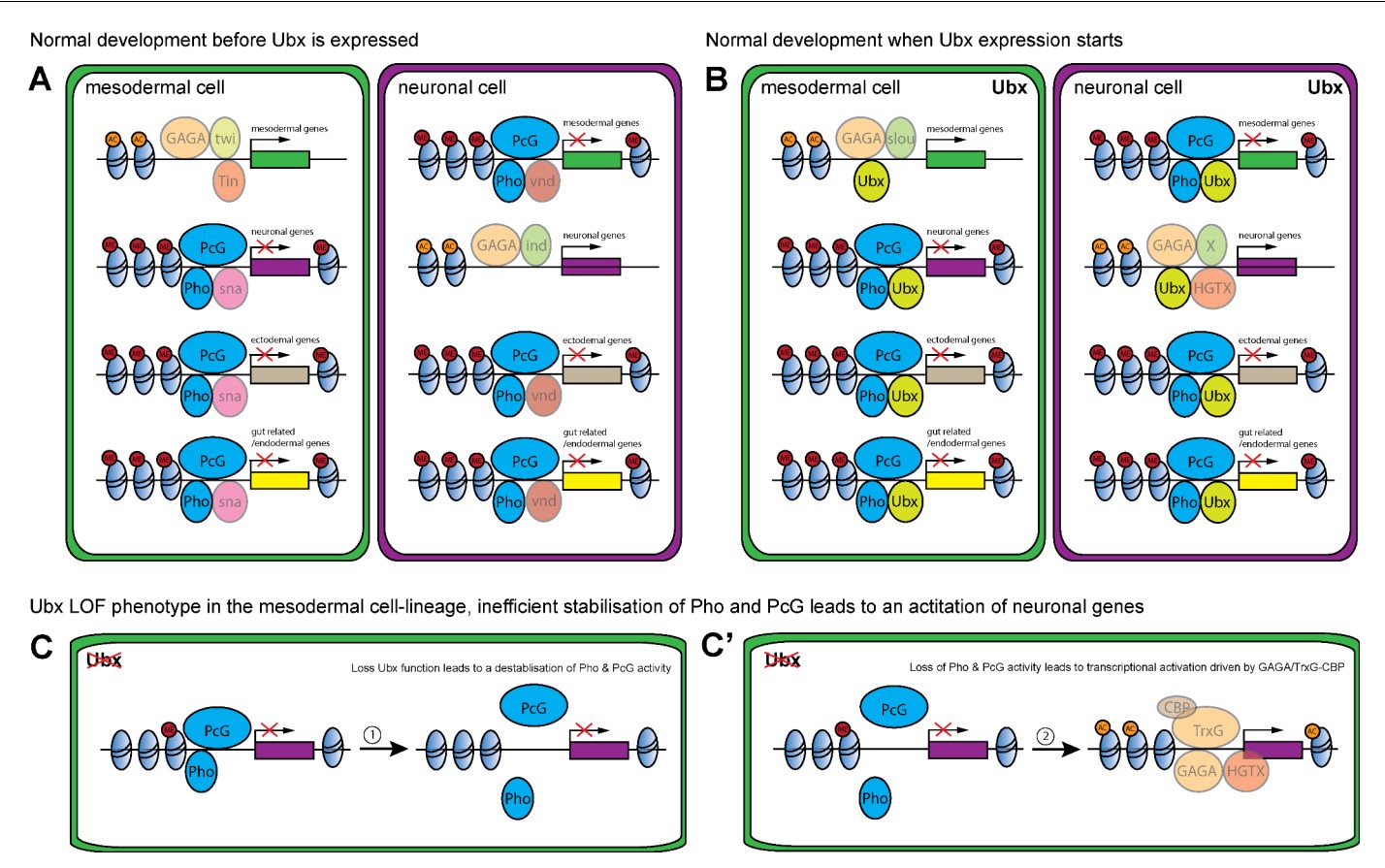

**Figure 8.** Model of Ubx and Pho combinatorial action in mesoderm development. (A, B) Chromatin regions close to genes encoding different lineage functions are shown in mesodermal (green) and neuronal (purple) wild-type cells before (A) and after (B) Ubx is expressed. (A) Mesodermal determination and specification factors, like Twist (Twi) and Tinman (Tin), activate mesodermal genes, possibly in combination with Trl (GAGA) and/or other factors. In the same cell, alternative fate genes (grey: ectodermal genes, yellow: gut related genes; purple: neuronal) are repressed, possibly by the mesoderm-specific TF Snail (sna) with Pho and PcG (dark blue). In neuronal determination and specification, TFs like Intermediate neuroblasts defective (Ind) activate neuronal genes, possibly in combination with with Trl (GAGA), while alternative fate genes (grey: ectodermal genes, yellow: gut related genes; green: mesodermal) are repressed, for example via Ventral nervous system defective (Vnd) and Pho and PcG (dark blue). (B) When Ubx expression starts, determination and specification factors are turned off or their expression gets restricted. Now, Ubx ensures the continued activation of the proper set of lineage-specific genes, while at the same time maintaining the repression of alternative fate genes. (C, C') Loss of Ubx in the mesodermal lineage destabilises Pho binding to regulatory regions of alternative fate genes (example for neuronal (purple)). (C) This results in a destabilisation of the PcG complex at these regions, leading to a loss/reduction of H3K23me3 marks. (C') In the absence of Ubx and PcG, the sites can be bound by the TrxG complex through Trl (GAGA) in combination with CBP. Consequently, Lys 27 of histone three at these regions will get acetylated, resulting in the activation of gene expression.

DOI: https://doi.org/10.7554/eLife.42675.015

The following figure supplement is available for figure 8:

**Figure supplement 1.** Antp is ectopically expressed in the mesoderm in the absence of Ubx.
DOI: https://doi.org/10.7554/eLife.42675.016

response elements for Trithorax group (TrxG) proteins (*Kingston and Tamkun, 2014*), a large group of proteins first described for their role in transcriptional activation. In this line, we identified not only the binding motif for Pho but also for the GAGA TF Trithorax-like (Trl) enriched among the Ubx bound and H3K27me3 marked chromatin sites, and we found Trl to interact with Ubx *in cellulo* (*Figure 8—figure supplement 1F*). Trl was initially discovered to activate transcription, in particular of the *Ubx* gene (*Biggin and Tjian, 1988*), but is also required for transcriptional repression, as it binds PREs (*Horard et al., 2000*) and physically associates with the Polycomb Repressive Complex 1 (*Poux et al., 2001*). Interestingly, Trl had been suggested recently to function as a pioneer factor in early *Drosophila* development by making genomic regions accessible through the deposition of active histone marks. Furthermore, it had been shown that another TrxG protein, the methyltransferase Trithorax (Trx), mediates together with the p300/CREB-binding protein (CBP) the deposition of H3K27ac marks (*Tie et al., 2016*; *Tie et al., 2009*), a histone mark that was enriched at promoters and enhancers upon Ubx depletion. Thus, the TrxG complex and CBP could be recruited in the absence of Ubx to activate gene expression (*Figure 8*). GAGA factors like Trl do not only play a role in the activation but is also in the repression of gene expression by interacting with the Polycomb complex (*Horard et al., 2000*; *Mulholland et al., 2003*). Interestingly, a recent study showed that GAGA factors are required for the formation of repressive chromatin loops in Polycomb domains to stabilize gene silencing during early *Drosophila* development (*Ogiyama et al., 2018*). These results could explain one puzzling result of our study, the increase of H3K27me3 marks in the absence of Ubx, which could be due to Trl interacting with these sites when Ubx levels drop (*Figure 8*), thereby promoting or increasing the formation of repressive chromatin loops. In future, it will be important to study TrxG as well as PcG proteins at Ubx targeted control regions to understand how Hox TFs orchestrate the interplay between transcriptional activation and repression in the course of lineage development when cellular plasticity needs to be restricted.

An important question arising from this study is how Hox TFs can function so specifically in different tissue lineages, as they activate a gene set in one lineage while repressing it in another one. One explanation is that tissue/lineage restricted factors and their interaction with Hox TFs provide specificity. Interestingly, we find the binding motif for the mesoderm-specific TF Snail (Sna) enriched among the sites, which are bound by Ubx and marked by H3K27me3 in the mesodermal lineage, while in the neuronal lineage the Ventral nervous system defective (Vnd) motif is frequently found in Ubx/H3K27me3 sites. Motifs for tissue-specific TFs are also detected at active sites, as Ubx/H3K27ac regions in the mesoderm have the Slouch (Slou) motif enriched, while in the nervous system it is the motif for Intermediate neuroblasts defective (Ind). Thus, we assume these factors to be critical in determining the lineage-specific output (*Figure 8*). In future, it will be interesting to study these tissue-restricted interactions and determine their role in the lineage-specific repression and activation of developmental gene programs.

One peculiar feature of Hox TFs is that they are not only active during the development but their input is continuously required throughout the lifetime of an organism to assess the positional values of cells and to maintain their proper identities and functions (*Argiropoulos and Humphries, 2007*; *Cerdá-Esteban and Spagnoli, 2014*; *Friedrich et al., 2016*; *Leucht et al., 2008*; *Morgan, 2006*; *Philippidou and Dasen, 2013*; *Philippidou et al., 2012*). Our findings now shed new light on the mechanism ensuring this stability in cellular identity, as they suggest that Hox TFs robustly repress alternative lineage programs (via the interaction with epigenetic factors like Pho) and reliably restrict the developmental potential encoded cells. In the normal context, this is absolutely critical for an organism to function properly, however, in some instances it can be a major hurdle, in particular when cells need or should adopt a new identity, which requires them often to regain plasticity. In organismal life, this happens mostly during dedifferentiation, regeneration and tissue remodelling (*Merrell and Stanger, 2016*; *Monier et al., 2005*; *Schaub et al., 2015*). Another situation of cell type conversion is cellular reprogramming, which is extensively studied due to its high potential as therapeutic strategy (*Lin et al., 2018*). But although reprogramming strategies have been improved over time, the direct conversion of one somatic cell type into another one, the so-called transdifferentiation, is still inefficient (*Ebert et al., 2015*; *Manandhar et al., 2017*). Interestingly, it has been reported recently that the induction of the Hox code typical for a differentiated cell type in pluripotent stem cells (PSCs), either in combination (*Sugimura et al., 2017*) with other factors or even alone (*Steens et al., 2017*), can convert PSCs into the cell type expressing the Hox code in vivo. This clearly shows that *Hox* genes can effectively induce cell type specific transcriptional programs in

multi-potent cells thereby unambiguously inducing their differentiation. This is in line with our findings showing that Hox TFs activate lineage-specific transcriptional profiles with high precision. However, so far it is unclear why the conversion of somatic cells of one type into another one is so difficult. Our data now suggest that one reason for this hurdle could be the Hox encoded restriction of cellular plasticity via the repression of alternative gene programs using Polycomb based epigenetic mechanisms.

## Materials and methods

### Fly stocks and husbandry

For the INTACT method (*Steiner et al., 2012*) the following fly lines were used: *twi-INTACT* and *UAS-INTACT* (gift from the Henikoff lab). The *UAS-INTACT* was crossed in the *elav-GAL4* background to generate *elav-GAL4;;UAS-INTACT* stable lines. For the degrade GFP experiments (*Caussinus et al., 2011*) the following lines were used: *arm-GAL4* (BL1561) *Mef2-GAL4* (BL50742), *UAS-TEV-P14* (homemade using the construct published in Taxis et al., 2009, insertions were performed by BestGene at attP5 and attP2). *UAS-Nslmb-vhhGFP4* (BL38419), the *GFP-Ubx*$^{3.005}$ line was recombined with the *UAS-Nslmb-vhhGFP4* and *Mef2-GAL4* lines and crossed in the *twi-INTACT* background to generate *twi-INTACT;Mef2-GAL4,GFP-Ubx*$^{3.005}$ and *twi-INTACT; UAS-Nslmb-vhhGFP4,GFP-Ubx*$^{3.005}$. Embryos from Ubx deGradFP experiments were collected at 10–18 hr AEL to ensure the knockdown of the GFP-Ubx protein. The *pho-like*$^{81A}$;*pho*$^1$ double mutant was generously provided by Jürg Müller (*Klymenko et al., 2006*). The *dpp-lacZ* line was obtained from Manfred Frasch.

### Generation of endogenously tagged Ubx

The *GFP-Ubx*$^{3.005}$ line was generating by using the CRIPR/Cas9 system (*Bassett et al., 2013*; *Gratz et al., 2013*). For the donor DNA a pUC-MCS-5'GFP-MCS was designed, multiple cloning sites flanking the GFP containing an ATP start codon. For the 5' region from the GFP a homologous arm (*Baena-Lopez et al., 2013*) containing the 5'UTR was cloned by using NotI and KpnI restriction sites. The 3' region from the GFP included the first Ubx exon and a large part of the intron for homologous recombination was cloned with BglII and XhoI. The gRNAs were designed to eliminate the first Ubx exon, positioned at the beginning of the 5'UTR and the end of the coding region of the first exon. The excised exon was replaced using the donor DNA and homologous recombination. The microinjection was performed by BestGene using *vas-Cas9* (BL51323) as injection line and the resulting progenies (F0) were crossed with TM3/TM6 balancers and resulting F1 was used for single crosses against TM3/TM6 balancers to generate independent stocks. The F1 generation was screened by PCR for the presence of the GFP and the GFP containing stocks of the F2 generation were visually screened for the Ubx patterned GFP expression in vivo.

### Generation of the Ubx antibody

The Ubx antibody was generated using the pGEX-purification system (GElifesciences). The open reading of Ubx-RA was cloned in the pGEX-6P-2 vector using BamHI and XhoI restriction site. The protein was purified according to the protocol (GElifesciences) and eluted by using the PreScission Protease site. The immunisation and antibody handling were performed by the Charles Rivers company.

### Purification of affinity-tagged nuclei, ChIP, ChIP-Seq and RNA-Seq

The nuclei were purified as described in *Steiner et al. (2012)*, the purification was optimised by using a larger magnet (20cm x 20cm x 10cm, holding 70 kg) and a homemade magnet holder (gift from C Schaub and M Frasch). For ChIP and ChIP-Seq experiments: Chromatin preparation and immune-precipitation were performed as described in *Sandmann et al. (2007)*. The following antibodies were used: H3K27ac (ab4729, Abcam), H3K27me3 (ab6002, Abcam), Rb-Pho$^{2-382}$ (generously provided by J. Müller) and the homemade guanine pig Ubx antibody (gp-Ubx). Total RNA was isolated with TRIZol (Invitrogen) and DNA digest was performed with the TURBO DNA-free Kit (Ambion). The material was analysis or validated by qPCR (Invitrogen Syber-Green-Mix) (*Supplementary files 2*, *3*). Genome-wide sequencing and material handling was performed in tight

cooperation with the Deep-Sequencing facility in Heidelberg. The library for genome-wide sequencing was prepared by using the ThruPLEX DNA-Seq Kit (Rubicon) for illumina sequencing.

## Bioinformatics analysis

### ChIP-Seq

1) Ubx binding events in wild-type nuclei: ChIP-seq experiments were performed at least in biological duplicates. 2) Pho binding, H3K27ac and H3K27me3 marks in Ubx degradation nuclei: ChIP-seq experiments were performed in biological triplicates (control, experiment). The reads were first analysed with FastQC and aligned with bowtie2 (`bowtie2 –very-sensitive -x Bowtie2Index/ genome sequence.txt -S output.sam` [*Langmead and Salzberg, 2012*]) against the dm6 Drosophila genome version using standard conditions, the peaks were called by using MACS2 under model-based broad settings (`macs2 callpeak -t IP-file.sam -c input-file.sam -f SAM -n < name > broad -g dm –keep-dup auto -B –broad-cutoff 0.1 3`). Wild-type reads were tested for their similarity by person correlation and the two most similar were combined using samtools for further analysis. The annotation of wild-type peaks was performed using ChIPseeker (*Yu et al., 2015*), ChIPpeakAnno (*Zhu et al., 2010*) and the Drosophila genome dm6. Annotations were not filtered for 'real' coding genes and all FBgnNumbers were used for further analysis. Identification of differential Histone and Pho Binding events were identified with DiffBind (*Ross-Innes et al., 2012*), statistical relevant regions were isolated with BEDtools (bedtools intersect; *Quinlan and Hall, 2010*). For GO-Term annotations and overrepresented GO-Term analysis was performed with the web-tools bioDBnet and PANTHER (GO biological function complete, Binomial, Bonferroi correction [*Thomas et al., 2003*]). Promoter and enhancer definition is related to *Shlyueva et al. (2016)* and *Erceg et al. (2017)*.

### Statistical analysis of specific regions

Regions of interest were isolated and bed fines generated. The regions were compared with results from DiffBind using bedtools intersect (*Quinlan and Hall, 2010*) or full reads of the control and Ubx-deGrad using bedtools coverage (*Quinlan and Hall, 2010*). Violin plots were calculated using R.

### Motif search

The motif search on defined regions was done by using the MEME-suite web-tool (*Frith et al., 2008*) or the command line (`meme-chip -db motif_databases/FLY/OnTheFly_2014_Drosophila.meme –meme-minw 6 –meme-maxw 20 –meme-nmotifs 100 –centrimo-local –centrimo-maxreg 250 seq.fa`). Enrichment of known motifs was performed with a AMT MEME tool (`ame –verbose 1 oc. –control shuffled_overlaps_onlyPho_ME_2.fa –bgformat 1 –scoring avg –method ranksum –pvalue-report-threshold 0.05 file.fa db/FLY/fly_factor_survey.meme db/FLY/idmmpmm2009.meme db/FLY/flyreg.v2.meme db/FLY/OnTheFly_2014_Drosophila.meme db/FLY/dmmpmm2009.meme`).

### Comparison of Ubx, Tin and Mef2

The Tin data set from *Jin et al. (2013)* and the Mef2 data set from *Sandmann et al. (2006)* was used, the Ubx data sets for the mesoderm were realigned against the Drosophila genome dm3 using bowtie and the peaks were newly called using MACS2.

### Analysis of Ubx and Pho ChIP-Seq data

The Pho data set was obtained from *Erceg et al. (2017)* and processed as decried above for Ubx against dm6. Genomic overlaps were performed with ChIPseeker (*Yu et al., 2015*) and ChIPpeakAnno (*Zhu et al., 2010*).

### RNA-Seq

1) wild-type: RNA-seq experiments were performed in biological duplicates. 2) Loss of Ubx function experiments: RNA-seq experiments were performed in biological triplicates. Obtained RNA reads were first analysed with FastQC and aligned with TopHat2 (*Kim et al., 2013*) (`tophat2 -p 8 G Drosophila_melanogaster.BDGP6.86.gtf -o output_folder_name BDGP6_86 converted_sequence.fastq`) against the BDGP6_86 Drosophila genome version using default settings

(BDGP5_75 for all comparisons against dm3). The count table for further analysis was generated using HTSeq (`htseq-count –f bam –r pos –m union –s no –t exon accepted_hits.bam Drosophila_melanogaster.BDGP6.86.gtf>counts.txt` [*Anders et al., 2015*]). The differential expression analysis was performed by using DESeq2 (*Love et al., 2014*). The overrepresented GO-Term were analysed with PANTHER (GO biological function complete, Binomial, Bonferroi correction). The general tissue-specific transcriptome was generated by using a general calculation of the RPKM (Equation: RPKM = (109 * number of reads mapped to a gene)/Total mapped reads in the experiment * exon length in base-pairs for a gene)). Genes containing a RPKM >5 were classified as transcribed according to (*Dezso et al., 2008*; *Kryuchkova-Mostacci and Robinson-Rechavi, 2016*).

## Visualisation of the genome-wide data

Preparation of the data for visualisation was performed with deeptools: `Normalization computeGCBias -b file_sorted.bam –effectiveGenomeSize 142573017 g dm6.2bit -l 250 –GCbiasFrequenciesFile file_freq.txt –biasPlot file_test_gc.pdf` and `correctGCBias -b file_sorted.bam –effectiveGenomeSize 142573017 g dm6.2bit –GCbiasFrequenciesFile file_freq.txt -o file_gc_corrected.bam`. The samples were normalized to the input to generate the bigwig file `bamCompare -b1 sample_gc_corrected.bam -b2 input_gc_corrected.bam -o file_S-I_100_subtract.bw –scaleFactorsMethod None –operation subtract –binSize 10 –effectiveGenomeSize 142573017 –normalizeUsing RPKM –smoothLength 100 –centerReads –extendReads 150`. The reads and annotations were visualised with IGV.

## Interactive data mining tool

The enrichment analysis method presented in this paper is implemented as a user-friendly Shiny web-application accessible via http://beta-weade.cos.uni-heidelberg.de. The user can select the set of genes to perform the GO enrichment analysis and the respective background independently. Results of the analysis are presented as a plot, an interactive table displaying significantly enriched GO groups, and an interactive heatmap, showing the counts of enriched GO terms within the respective higher-order GO group. It is also possible to get an insight into the individual GO terms that make up a category and into the genes that contributed to the categories or terms. The functionality of the tool exceeds what is described here, a detailed documentation of the tool is deposited under http://beta-weade.cos.uni-heidelberg.de and additionally an interactive guide is provided in the online application.

## Multiple testing of GO-terms

Pre-set list of GO-terms related to mesoderm and muscle development and function (mesoderm), neuronal and central nervous system (CNS) and neuromuscular junction development and function (neuronal), ectoderm and trachea development and function (ectoderm) and endoderm development and gut structures (salivary glands, Malpighian tubes, hind- and foregut) and function (gut and endoderm) was generated (*Supplementary file 4*). Genes associated with different categories were analysed using an R program based on a biomaRt script (*Durinck et al., 2005*).

## Immunofluorescence staining and microscopy

### Nuclei staining

INTACT-sorted nuclei were collected by centrifugation at 1000 g for 5 min and resuspend in the suitable amount of HB125 (*Steiner et al., 2012*) +5% BSA for the antibody staining according to the conditions of the 1[st] antibodies and incubated over night at 4°C, than washed 3 times 15 min with HB125 and incubated in 500 µl HB125 +5% BSA for the 2[nd] antibodies (2[nd] ABs 1:500). Before mounting, nuclei were washed 3 times 15 min with HB125. Embryonic stainings were performed as described in *Domsch et al. (2013)* and the following antibodies were used: Rb-Mef2 (1:1500, Gift from H. Nguyen), Rat-elav (1:50, DSHB, ELAV-9F8A9), Rat-Tm1 (1:200, MAC-141, ab50567, Abcam), Rb-GFP (1:300, Invitrogen), Rat-GFP (1:100, ChromoTek), DAPI (1:500, Invitrogen), Rat-Org1 (1:100) (*Schaub et al., 2015*), Rb-Pho (1:250) (*Klymenko et al., 2006*), Ptx1 antibody (1:1000) (*Vorbrüggen et al., 1997*), ms-Antp (1:100, DSHB, 8C11). Amplification was obtained with the TSA system (Perkin-Elmer). Biotinylated (1:200, Vector Labs) and fluorescent (1:200, Jackson

ImmunoResearch) secondary antibodies were used. Images were acquired on the Leica SP9 Microscope using a standard 20x and 63x objective. The collected images were analysed and processed with the Leica program and Adobe Photoshop.

## Quantifications of data
### Quantification of the GFP signal intensity
GFP signal was measured with the Fiji program from at least five segments per embryo in five embryos. The mean and the t-test for significance were calculated in Excel.

### Quantification of histone marks
Regions of interest (Ubx bound regions with histone mark changes that are up-regulated or without up-regulation, Ubx and Pho bound regions, Ubx Pho and active regions) were identified and compared with the deGrad-DiffBind results, histone binding peaks from control or deGrad-data sets by using bedtools intersect or bedtools coverage. The isolated Fold or peak depth were illustrated in a Violin plot with R, as well as the wilcoxon rank test for significance. Fold changes required for ratio calculation were processed with an R-script (upon request) and illustrated in a Violin plot with R, as well as the Wilcoxon rank test for significance.

## Protein IP, CoIP and Western Blot
### Embryonic Protein IP
The GFP-Ubx protein was immune precipitated from whole mount embryos by using GFP-tap-beads (ChromoTek). Collected Embryos were washed and smashed in IP Buffer (50 mM Tris-HCl, pH 8.0; 1 mM EDTA; 0.5% (v/v) Triton X-100, 1 mM PMSF, Complete proteinase inhibitors (ROCHE). For immunoprecipitation 1 mg of total protein was incubated over night with IP Buffer washed GFP-tap-beads. Beads were then washed 3x with IP Buffer and resuspended in Laemmli buffer.

### Cell culture CoIP
S2R + Drosophila cells were cultured in Schneider medium supplemented with 10% FCS, 10 U/ml penicillin and 10 µg/ml streptomycin. For plasmid transfections, $10.10^6$ cells per ml were seeded in 100 mm dishes and transfected with Effectene (Qiagen) according to the manufacturer's protocol. Cells were harvested in Phosphate Buffered Saline (PBS) and pellets were resuspended in NP40 buffer (20 mM Tris pH7.5, 150 mM NaCl, 2 mM EDTA, 1% NP40) supplemented with protease inhibitor cocktail (Sigma-Aldrich) and 1 mM of DTT. For co-immunoprecipitation assays 1 mg to 1.5 mg of proteins were incubated for 3 hr with 15 µl of GFP-Trap beads (Chromotek). Beads were then washed 3 times with NP40 buffer and finally resuspended in Laemmli buffer for immunoblotting analysis. Input fractions represent 2.5% of the immunoprecipitated fraction.

### Embryonic CoIP
Overnight collection of embryos was dechorionated, fixed with 3.2% formaldehyde and collected in Phosphate Buffered Saline (PBS) supplemented with Tween 0.1%. Pellets were resuspended in buffer A (10 mM Hepes pH 7.9, 10 mM KCl, 1.5 mM MgCl2, 0.34 M sucrose, 10% glycerol) and dounced 25–30 times with loose- and 5 times with tight- pestle. Lysates were incubated with 0.1% Triton and centrifugated. Nuclear pellets were then resuspended with buffer B (3 mM EDTA pH8, 0.2 mM EGTA pH8), sonicated (Picoruptor, Diagenode), and treated with Benzonase (Sigma). For co-immunoprecipitation assays 4 to 5 mg of nuclear lysates were diluted in NP40 buffer (20 mM Tris pH7.5, 150 mM NaCl, 2 mM EDTA, 1% NP40) and incubated overnight with 40 µl of GFP-Trap beads (Chromotek). Beads were then washed 5 times with NP40 buffer and resuspended in Laemmli buffer for immunoblotting analysis. All buffers were supplemented with protease inhibitor cocktail (Sigma) and 1 mM of DTT and 0.1 mM PMSF. Input fractions represent 2.5% of the immunoprecipitated fraction.

For western blot analysis, proteins were resolved on 10% SDS-PAGE, blotted onto PVDF membrane (Biorad) and probed with specific antibodies after saturation. The antibodies (and their dilution) used in this study were: Ubx (home-made, 1/200), Pho (generously provided by Jürg Müller, 1/500 [*Klymenko et al., 2006*]), Histone 3 (ab1791, Abcam, 1/10000), GFP (A-11122, Life Technologies, 1/3000).

## Acknowledgements

We would like to thank the *Drosophila* community for providing us generously with fly stocks and antibodies, in particular Steven Henikoff for the *twi-INTACT* and *UAS-INTACT* fly lines, Jürg Müller for the Pho antibody and the *phol*[81A];*pho*[1] double mutants and Gerd Vorbrüggen for the Ptx-1 antibody. Manfred Frasch and Christoph Schaub at the University of Erlangen-Nuremberg in Germany for the improved INTACT apparatus and the org-1 antibody. David Ibberson from the Heidelberg Deep Sequencing Core facility at Heidelberg University and EMBL for handling and supporting the genome-wide sequencing procedure, and Eugen Rempel for the Bioinformatics support. We also thank Lazaro Centanin, Gergana Dobreva, Stefan Stricker and all the members of the lab for critically reading the manuscript. We apologize to all whose work was not cited due to space limitations. This work was supported by the DFG (DFG, LO 844/8–1).

## Additional information

### Funding

| Funder | Grant reference number | Author |
| --- | --- | --- |
| Deutsche Forschungsgemeinschaft | LO 844 8/1 | Ingrid Lohmann |

The funders had no role in study design, data collection and interpretation, or the decision to submit the work for publication.

### Author contributions

Katrin Domsch, Conceptualization, Data curation, Formal analysis, Validation, Investigation, Visualization, Methodology, Writing—original draft, Writing—review and editing; Julie Carnesecchi, Data curation, Formal analysis, Validation, Visualization, Methodology, Writing—original draft, Writing—review and editing; Vanessa Disela, Jana Friedrich, Olga Ermakova, Maria Polychronidou, Data curation, Investigation, Methodology, Writing—review and editing; Nils Trost, Data curation, Software, Investigation, Methodology, Writing—review and editing; Ingrid Lohmann, Conceptualization, Formal analysis, Supervision, Funding acquisition, Visualization, Writing—original draft, Project administration, Writing—review and editing

### Author ORCIDs

Nils Trost http://orcid.org/0000-0002-5171-018X
Ingrid Lohmann http://orcid.org/0000-0002-0918-2758

### Decision letter and Author response

Decision letter https://doi.org/10.7554/eLife.42675.033
Author response https://doi.org/10.7554/eLife.42675.034

## Additional files

### Supplementary files

• Supplementary file 1. Table of known genes expressed in the mesodermal or neuronal lineages.
DOI: https://doi.org/10.7554/eLife.42675.017

• Supplementary file 2. Primers used for qPCR experiments to test total RNA.
DOI: https://doi.org/10.7554/eLife.42675.018

• Supplementary file 3. Primers used for qPCR to test genomic loci identified by ChIP experiments.
DOI: https://doi.org/10.7554/eLife.42675.019

• Supplementary file 4. Collection of GO terms grouped into different categories for multiple GO term testing.
DOI: https://doi.org/10.7554/eLife.42675.020

• Transparent reporting form

DOI: https://doi.org/10.7554/eLife.42675.021

## Data availability

Sequencing data have been deposited in GEO: GSE121754 (all), GSE121670 (RNA-Seq) & GSE121752 (ChIP-Seq).

The following datasets were generated:

| Author(s) | Year | Dataset title | Dataset URL | Database and Identifier |
|---|---|---|---|---|
| Katrin Domsch, Julie Carnesecchi, Vanessa Disela, Jana Friedrich, Nils Trost, Olga Ermakova, Maria Polychronidou, Ingrid Lohmann | 2019 | All sequencing data for 'The Hox transcription factor Ubx stabilizes lineage commitment by suppressing cellular plasticity in Drosophila' | https://www.ncbi.nlm.nih.gov/geo/query/acc.cgi?acc=GSE121754 | NCBI Gene Expression Omnibus, GSE121754 |
| Katrin Domsch, Julie Carnesecchi, Vanessa Disela, Jana Friedrich, Nils Trost, Olga Ermakova, Maria Polychronidou, Ingrid Lohmann | 2019 | The Hox Transcription Factor Ubx stabilizes Lineage Commitment by Suppressing Cellular Plasticity [RNA-seq] | https://www.ncbi.nlm.nih.gov/geo/query/acc.cgi?acc=GSE121670 | NCBI Gene Expression Omnibus, GSE121670 |
| Katrin Domsch, Julie Carnesecchi, Vanessa Disela, Jana Friedrich, Nils Trost, Olga Ermakova, Maria Polychronidou, Ingrid Lohmann | 2019 | The Hox Transcription Factor Ubx stabilizes Lineage Commitment by Suppressing Cellular Plasticity [ChIP-seq] | https://www.ncbi.nlm.nih.gov/geo/query/acc.cgi?acc=GSE121752 | NCBI Gene Expression Omnibus, GSE121752 |

The following previously published datasets were used:

| Author(s) | Year | Dataset title | Dataset URL | Database and Identifier |
|---|---|---|---|---|
| Erceg, Pakozdi, Marco-Ferreres, Ghavi-Helm, Girardot, Bracken, Furlong | 2017 | Time course Pho/dSfmbt Chromatin Immunoprecipitation on Drosophila melanogaster embryos during embryogenesis | http://www.ebi.ac.uk/ena/data/view/ERR1358767-ERR1358777 | European Nucleotide Archive, ERR1358767-ERR1358777 |
| Jin H, Stojnic R, Adryan B, Ozdemir A, Stathopoulos A, Frasch M | 2013 | Genome-wide Tinman binding sites in early and late Drosophila embryos | https://www.ncbi.nlm.nih.gov/geo/query/acc.cgi?acc=GSE41628 | NCBI Gene Expression Omnibus, GSE41628 |

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
