## [Decision Letter]

Thank you for submitting your article "The Hox Transcription Factor Ubx stabilizes Lineage Commitment by Suppressing Cellular Plasticity" for consideration by *eLife*. Your article has been reviewed by three peer reviewers, and the evaluation has been overseen by a Reviewing Editor and Kevin Struhl as the Senior Editor. The following individual involved in review of your submission has agreed to reveal his identity: Bart Deplancke (Reviewer #2).

The reviewers have discussed the reviews with one another, at length, and the Reviewing Editor has drafted this decision to help you prepare a revised submission.

Overall, the reviewers were enthusiastic about the subject and the approach, and they found the conclusions interesting and noteworthy. But as you can see from the reviews, which are appended in their entirety below, the reviewers also had substantial reservations about technical aspects of data collection and analysis. One simple issue that all the reviewers were concerned about was the lack of replicates for the ChIP experiments (3 replicates are standard), and they agreed that 2 or preferably 3 replicates should be performed and the data re-analyzed. A second general concern was the ChIP-PCR data indicating that Ubx controls H3 K27me3 distributions and Pho binding; in this case the reviewers agreed that the Pho ChIP samples +/- Ubx-deGrad (Figure 6) should be sequenced, to better substantiate your interesting conclusions using whole genome data. (In principle, the H3 K27me3 ChIP samples in Figure 6 should likewise be sequenced, though we leave it up to you to decide how important it is.) Thirdly, we would like to see a few changes to the data analysis, as described in the reviews, to bring into line with standards in the field. Clearly these additional experiments will require some investment, but we feel they are necessary to the bring to paper up to accepted standards in the field. We look forward to seeing your revision.

*Reviewer #1:*

The Lohmann lab presents an extremely complex analysis of the role of the Hox transcription factor Ubx in lineage commitment/specification in *Drosophila*. In order to remove Ubx from a given lineage only, they used a protein degradation method recently established and an endogenously tagged version of Ubx. The study is elegant in its design, and many of the discussed findings appear to provide interesting insight into the role of Ubx in the mesoderm, and possibly beyond.

However, for a non-expert in such genome-wide approaches, both the experimental and the analytical part of the work are difficult to analyse and evaluate in detail. In light of this fact, I have to admit that I found the work very interesting, timely and novel, and certainly of excellent quality and relevance to be published in *eLife*. Even after having read the paper three times, I have not seen any reason to ask for additional experiments.

*Reviewer #2:*

Using the INTACT nuclei isolation method coupled to comprehensive RNA-seq and ChIP-seq assays, Domsch and colleagues found that a Hox class transcription factor (TF) Ubx binds to both active genes and inactive genes linked to either the mesodermal or neuronal lineages. The authors therefore hypothesized that Ubx may function in tissue development as a repressor of alternative transcriptional programs (i.e. programs steering cells toward a distinct differentiation fate). To investigate this postulate, the authors studied the regulatory function of Ubx in different lineages by depleting this TF in a tissue-specific manner. These experiments revealed that a large proportion of alternate fate genes are upregulated once Ubx is removed, in line with their hypothesis. The authors further profiled the H3K27ac and H3K27me3 chromatin landscapes in Ubx-depleted tissues, which revealed that Ubx likely represses alternate fate genes by controlling their epigenomic status, especially K3K27me3 enrichment. Subsequent motif analysis and downstream experiments thereby showed that this repressive chromatin mark may be directed toward these loci through the regulator Pho, which is stabilized by Ubx at these repressed regions.

Together, this is a very interesting study that should be of broad interest to the gene regulation and developmental biology fields given the formulation of a novel mechanistic model that rationalizes how a broadly expressed TF is able to contribute to cell lineage specification by repressing alternate cell fate genes. The techniques applied, such as tissue-specific nuclei isolation, tissue specific Ubx gene tagging/depletion are elegant and state of the art and the resulting data seems in general solid. There are only a few items that could be clarified better:

1) The authors used 10 RPKM as their threshold to distinguish active from inactive genes. This is a rather high threshold, and so one is left to wonder what the consequence would be on the downstream analyses/results if the more standard threshold of 1 RPKM would be applied?

2) The authors show that Ubx binds to a very large number of genes (roughly 10 K based on Figure 2B, and thus >70% of all *Drosophila* (10 K / 14 K) can be considered to be Ubx targets). This raises a couple of questions:

a) In Figure 2B, the number of Ubx-bound genes in neuronal tissue at stages 14-17 is much lower than for the other three conditions. What could be the underlying reason?

b) In Figure 2G, the authors show that there is a high correlation in higher order GO terms between the lineage specific transcriptional and Ubx binding profiles. As a control, the authors included the TF Tinman, which revealed a more distinct GO term enrichment profile. However, since Ubx binds to such a large number of genes, is it not entirely expected that the majority of active genes will also be Ubx targets, and thus yield the same GO term enrichment profile? The same holds true for the inactive genes which also fall mostly within the "Ubx target space" and would thus again yield the same profile. That Tin does not yield the same profile is also not surprising given that it binds much fewer genes so the overlap between (in)active genes and Tin gene targets will be much smaller. A better control (if available) would in this regard be a TF that, similar to Ubx, binds to a large number (10 K) of genes. In its current configuration however, this analysis is not very informative. Similar concerns apply to the statements "Strikingly, 85% (1227/1452) of the genes with reduced and 90% (1299/1393) of the genes with increased expression were bound by Ubx in mesodermal nuclei in wild-type embryos, implying that most of the expression changes were a direct consequence of altered Ubx chromatin interactions". Again, since >70% of *Drosophila* genes are Ubx targets, 85-90% is in fact not very striking and could just be a statistical aberration. This needs to be addressed.

3) The proposed model, while informative, is too simplistic. This is because, in its current form, the model lacks information on what the factors / determinants are that mediate either tissue-specific activation or repression. The identification of these factors is likely beyond the scope of the current study, but perhaps some first layer data analysis on motifs (beyond the listed generic ones) could be performed. In addition, these additional layers need to be incorporated in the model because, as an example, the Ubx-Pho complex is shown to repress neuronal genes in the mesoderm given the enrichment of the Pho motif in these repressed genes. But why then, given that the same motif will obviously be present in the DNA of a neuronal cell, does Pho-Ubx not repress these same genes in a neuronal cell (also since both Pho and Ubx are ubiquitously expressed)? What makes that the Pho motif in a neuronal cell is read differently in a mesodermal cell and vice versa? The same reasoning can be followed for active genes in the respective cell types. At a minimum, such uncertainties should be represented in the model to clearly indicate what type of regulatory information is still missing.

*Reviewer #3:*

I have mixed responses to the paper by Domsch et al. On the one hand, the experimental design, to analyze Hox regulatory mechanisms in distinct embryonic tissues, seems at first glance important and exciting, I was less impressed with the analysis than I expected to be. My concerns fall in two categories:

1) Data generation and analysis.

For one, although replicates were analyzed for the RNA-seq experiments, it seems that all of the ChIP-seq datasets were only done once, without replicates. Given how noisy these type of data can be (especially when working with a small number of sorted cells and weak ChIP signals as in this case) I think it is imperative to compare at least two experimental replicates and use peak calling algorithms that require two independent datasets to assess statistical relevance. The Venn diagrams in Figure 2—figure supplement 1 for example suggest that there are ~14,000 (!) Ubx peaks in the mesoderm and that more than 7000 of these are mesoderm-specific. These numbers are quite striking but as it stands it is not possible to know how solid they are without replicate datasets and rigorous statistical tests.

I also wonder about the way the authors subtract input ChIP signals from their experimental IPs, as this is an analysis method that is not typically seen in the literature. I would like to see how noisy the input IPs are -- those tracks should be shown.

There is little validation of the ChIP peaks. The only known enhancer that was highlighted was one from Dpp.

The Pho co-IP signal is surprisingly weak given the mechanism of co-recruitment that the authors are proposing.

GAGA-like motifs are often observed by motif discovery algorithms and I am not convinced they are meaningful here given the lack of solid follow up evidence.

One potential strength of the paper is the claim that Pho binding and K27me3 depends on Ubx binding. However, this conclusion is based on a small handful of loci analyzed by qPCR and the differences are small. Further, given that the Pc system can have very specific targets, I think that calculating the ratio of K27me3/K27ac can be misleading. Better to treat these marks independently, since when one is lower it isn't necessarily the case that the other will be higher. It is also the case that Pho binding is not equivalent to K27me3; the latter could simply be a consequence of whether the gene is repressed, which may or may not occur in a Pho-dependent manner.

Many genes lose expression in the Ubx knockdown experiments, suggesting that Ubx also acts as an activator in the mesoderm, but this is not adequately discussed or studied in the paper. It is not clear why the authors choose to focus on the repressed genes.

2) General comments.

Ubx, and Hox genes in general, are well studied transcription factors that are known to be important in multiple cell types including mesoderm, the nervous system, and epithelium. So, seeing lots of genes change their expression when a broadly used transcription factor like Ubx is knocked down. It is also not surprising that the genes differ depending on the tissue (thus the GO analyses done in this paper seems rather trivial). What would take the analyses beyond what is known already is to understand how Ubx carries out differential functions in these two (and other) tissues. Presumably this happens in conjunction with tissue specific factors (e.g. twist for the mesoderm). The role of lineage specific factors, however, is not addressed and only marginally mentioned in the paper.

Terms such as "Cell plasticity" and "lineage restriction" are used rather loosely. The authors show that many up-regulated genes after Ubx knockdown are expressed in other lineages, but this result would have to be the case when a TF that represses genes is removed. There is no hard-evidence of lineage conversion (like loss of mesodermal master-regulatory factors for example), nor is it clear why neuronal (as opposed to endodermal or epithelial genes) genes are the focus here. It seems that the analysis is artificially restricted to repressed neuronal genes, while a more unbiased analysis of the changes that occur would be more valuable.

---

## [Author Response]

Overall, the reviewers were enthusiastic about the subject and the approach, and they found the conclusions interesting and noteworthy. But as you can see from the reviews, which are appended in their entirety below, the reviewers also had substantial reservations about technical aspects of data collection and analysis.One simple issue that all the reviewers were concerned about was the lack of replicates for the ChIP experiments (3 replicates are standard), and they agreed that 2 or preferably 3 replicates should be performed and the data re-analyzed.

We apologize if this was written in an unclear manner, but we had done all experiments of course at least in duplicates. We have now redone the analysis for the Ubx ChIP-seq data, all replicates were tested for similarity by Pearson correlation (0.94 – 0.96) and we used two replicates with highest similarity for further analysis. This is now clearly written in the revised version.

The experiments in the Ubx degradation background were performed in three biological replicated, for the control (twi-INTACT; GFP-Ubx, UAS-Nslmb) and the experiment (twi-INTACT; *Mef2*>UAS-Nslmb,GFP-Ubx). For the analysis of differential Expression (RNA-Seq, DESeq) and differential Binding (ChIP-seq, DiffBind) the two most similar replicates were analyzed.

We emphasize this now very clearly in the main text as well as in the Materials and methods section.

A second general concern was the ChIP-PCR data indicating that Ubx controls H3 K27me3 distributions and Pho binding; in this case the reviewers agreed that the Pho ChIP samples +/- Ubx-deGrad (Figure 6) should be sequenced, to better substantiate your interesting conclusions using whole genome data. (In principle, the H3 K27me3 ChIP samples in Figure 6 should likewise be sequenced, though we leave it up to you to decide how important it is.)

We have performed the requested Pho ChIP experiments in the UbxDegrad and control and sequenced everything in three biological replicates. The data is now included in the manuscript and shown in Figure 7.

Thirdly, we would like to see a few changes to the data analysis, as described in the reviews, to bring into line with standards in the field.

We have performed the requested analyses and describe them in the specific points to the reviewers (in particular reviewer #3).

Reviewer #2:

[…] 1) The authors used 10 RPKM as their threshold to distinguish active from inactive genes. This is a rather high threshold, and so one is left to wonder what the consequence would be on the downstream analyses/results if the more standard threshold of 1 RPKM would be applied?

We thank the reviewer for this comment. The usage of such a stringent threshold was based on previous studies, which had used similar RPKMs, in particular when studying tissue-specific gene expression (for example Dezso Z, … Nikolskaya, BMC Biology, 2008). In addition, other studies using the INTACT procedure and thus profiling nuclear and not whole cell RNAs used a similar stringent threshold to reduce the false positive recovery rate (for example, Reynoso et al., 2019). We do understand the reviewer’s concerns, thus we have re-analyzed the data using a threshold of >5 RPKM, which is frequently used (Dezso et al., 2008, Chen et al., 2013, reviewed Kryuchkova-Mostacci and Robinson-Rechavi, 2016). As the reviewer will appreciate, this does not change the results and the main take home message of the manuscript.

2) The authors show that Ubx binds to a very large number of genes (roughly 10 K based on Figure 2B, and thus >70% of all Drosophila (10 K / 14 K) can be considered to be Ubx targets). This raises a couple of questions:a) In Figure 2B, the number of Ubx-bound genes in neuronal tissue at stages 14-17 is much lower than for the other three conditions. What could be the underlying reason?b) In Figure 2G, the authors show that there is a high correlation in higher order GO terms between the lineage specific transcriptional and Ubx binding profiles. As a control, the authors included the TF Tinman, which revealed a more distinct GO term enrichment profile. However, since Ubx binds to such a large number of genes, is it not entirely expected that the majority of active genes will also be Ubx targets, and thus yield the same GO term enrichment profile? The same holds true for the inactive genes which also fall mostly within the "Ubx target space" and would thus again yield the same profile. That Tin does not yield the same profile is also not surprising given that it binds much fewer genes so the overlap between (in)active genes and Tin gene targets will be much smaller. A better control (if available) would in this regard be a TF that, similar to Ubx, binds to a large number (10 K) of genes. In its current configuration however, this analysis is not very informative. Similar concerns apply to the statements "Strikingly, 85% (1227/1452) of the genes with reduced and 90% (1299/1393) of the genes with increased expression were bound by Ubx in mesodermal nuclei in wild-type embryos, implying that most of the expression changes were a direct consequence of altered Ubx chromatin interactions". Again, since >70% of Drosophila genes are Ubx targets, 85-90% is in fact not very striking and could just be a statistical aberration. This needs to be addressed.

We thank the reviewer again for this valuable comment. Concerning the first point: We cannot explain why Ubx binds to less chromatin regions in the late neuronal time point, in particular as the quality of the reads is similar to the other stages. However, we assume that this is due to the different biology of the two lineages. In particular, as we already mention in the text, the neuronal lineage is a bit ahead of the mesodermal lineage with regards to developmental processes and initiates differentiation earlier than the mesoderm. This might result in a reduction of Ubx binding in the neuronal lineage.

The surprising outcome was that Ubx is indeed binding to a lot of sites. This is of course dependent on the TF type, on the way data are analysed, which thresholds are set. As requested by this reviewer before, we reanalyzed the Ubx data sets using a threshold of >5 RPKM (dm6 *Drosophila* genome), we then identified matching replicated via person correlation and combined the data sets for further analysis. This already changed the number of genes to which Ubx binds. We obtain now 6417 Ubx bound genes at stages 10-13 and 8256 Ubx bound genes at stages 14-17 in the mesoderm, with 80% of them corresponding to coding genes and 20% to non-coding genes. Thus, in total Ubx binds to 37-46% of the genes in the *Drosophila* genome (total number of coding genes in the dm6 genome is 13971) using these settings. Using the WEADE tool, we still see the same result, that Ubx binding reflects genome activity, suggesting that Ubx indeed comprehensively controls lineage development.

We have studied the comparison of Ubx and Tin binding behavior in more detail, as we do agree with the reviewer that the number of bound genes will affect the outcome. For Tin, we had to rely on available mapping data onto the *Drosophila* genome dm3, since the files (bed files) available for Tin (Jin et al., 2013) were done using this version. Alignment and peak calling against dm3 results in regions that are named “U” and “Uextra”, which are normally removed as described in the supplementary of Erceg et al., 2017. This led to a removal of nearly half of the mapped reads. Using dm6 as a reference genome, which we did for the Ubx data, maintains these peaks and results in more binding events. In order to make the Ubx and Tin datasets more comparable, we mapped the Ubx peaks against the dm3 genome, which resulted in the identification of only 1914 genes to which Ubx binds in the early mesoderm. This is now comparable to the binding events of Tin in the mesoderm. We then redid the WEADE analysis and still get a very similar result (which we now present in Figure 2—figure supplement 2). In addition, we did the analysis also with another TF, *Mef2*, which generally controls mesoderm development and not like Tin a sub-aspect (cardiac development). Interestingly, we see that Ubx and Medf2 binding profiles are similar, while Tin binding preferences are more different. This makes in our opinion the main message even stronger, as Ubx binding is more similar to a general factor controlling mesoderm development. Thus, we are convinced that this finding reflects true differences between Ubx, *Mef2* and Tin.

We do agree with the reviewer, that in the previous version the use of “striking” was not adequate, and we removed it from the text. Using the reanalyzed data, we now see that about 70% of the genes that were up- or down-regulated were bound by Ubx, and thus are very likely direct targets of Ubx. We would however highlight that binding does not necessarily mean a regulatory function, it is possible that Hox proteins bind without any functional consequences, and this could be a substantial fraction of the Hox binding events (see Biggin and McGinnis, 1997).

3) The proposed model, while informative, is too simplistic. This is because, in its current form, the model lacks information on what the factors / determinants are that mediate either tissue-specific activation or repression. The identification of these factors is likely beyond the scope of the current study, but perhaps some first layer data analysis on motifs (beyond the listed generic ones) could be performed. In addition, these additional layers need to be incorporated in the model because, as an example, the Ubx-Pho complex is shown to repress neuronal genes in the mesoderm given the enrichment of the Pho motif in these repressed genes. But why then, given that the same motif will obviously be present in the DNA of a neuronal cell, does Pho-Ubx not repress these same genes in a neuronal cell (also since both Pho and Ubx are ubiquitously expressed)? What makes that the Pho motif in a neuronal cell is read differently in a mesodermal cell and vice versa? The same reasoning can be followed for active genes in the respective cell types. At a minimum, such uncertainties should be represented in the model to clearly indicate what type of regulatory information is still missing.

We do agree with the reviewer that the factors that mediate the tissue-specific activation and repression are very interesting, but are indeed beyond the scope of this paper. We have however performed some preliminary analysis (motif search) and included this in an extended version of the model shown in Figure 8A and B. We have also now dedicated a section to the Discussion part where we explain more in depth how we envision that specificity in tissue-specific target gene regulation by Hox TFs is achieved.

Reviewer #3:

I have mixed responses to the paper by Domsch et al. On the one hand, the experimental design, to analyze Hox regulatory mechanisms in distinct embryonic tissues, seems at first glance important and exciting, I was less impressed with the analysis than I expected to be. My concerns fall in two categories:1) Data generation and analysis.For one, although replicates were analyzed for the RNA-seq experiments, it seems that all of the ChIP-seq datasets were only done once, without replicates. Given how noisy these type of data can be (especially when working with a small number of sorted cells and weak ChIP signals as in this case) I think it is imperative to compare at least two experimental replicates and use peak calling algorithms that require two independent datasets to assess statistical relevance. The Venn diagrams in Figure 2—figure supplement 1 for example suggest that there are ~14,000 (!) Ubx peaks in the mesoderm and that more than 7000 of these are mesoderm-specific. These numbers are quite striking but as it stands it is not possible to know how solid they are without replicate datasets and rigorous statistical tests.

We apologize if this was written in an unclear manner, but we had done all experiments of course at least in duplicates. To clarify: for the general mapping of tissue specific binding sites of Ubx we performed ChIP and HiSeq on two biological replicated. The experiments of the Ubx degradation (UbxDegrad) were performed in three biological replicates, the control (twi-INTACT; GFP-Ubx, UAS-Nslmb) as well as the experiment (twi-INTACT; *Mef2*>UAS-Nslmb,GFP-Ubx). For the analysis of differential Expression (RNA-Seq, DESeq) and differential Binding (ChIP-seq, DiffBind) the two most similar replicates were identified and analyzed. For the identification of active genes or Ubx bound regions, we identified matching Ubx ChIP-seq replicates via person correlation, combined the data sets for further analysis and have (as requested by reviewer #2) used a less stringent threshold of >5 RPKM threshold for the tissue specific transcriptome data sets (dm6 *Drosophila* genome). This changed the number of genes to which Ubx binds. We obtain now 6417 Ubx bound genes at stages 10-13 and 8256 Ubx bound genes at stages 14-17 in the mesoderm, with 80% of them corresponding to coding genes and 20% to non-coding genes. Thus, in total Ubx binds to 37-46% of the genes in the *Drosophila* genome (total number of coding genes in the dm6 genome is 13971) using these settings. Although this number is now reduced, the overall take home message is the same: Ubx binds a lot of sites, Ubx controls lineage development comprehensively (see comments to reviewer #2), and Ubx controls lineage development by repressing alternative fate genes and by activating lineage specific genes.

I also wonder about the way the authors subtract input ChIP signals from their experimental IPs, as this is an analysis method that is not typically seen in the literature. I would like to see how noisy the input IPs are -- those tracks should be shown.

As requested by the reviewer, we have added the input tracts to one sample (Figure 2—figure supplement 1A). We would however highlight that the subtraction of the Input from the IP is a common method to show the binding peaks and is used by many researcher groups, as it is normalizing the ChIP-seq reads (Erceg et al., 2017; Gaertner et al., 2012).

There is little validation of the ChIP peaks. The only known enhancer that was highlighted was one from Dpp.

As requested by the reviewer, we now show another example in Figure 4—figure supplement 1A, the *βTub60D* enhancer, this gene is significantly down-regulated upon Ubx degradation.

The Pho co-IP signal is surprisingly weak given the mechanism of co-recruitment that the authors are proposing.

We have performed whole embryonic Co-IP using the GFP-Ubx line, as a control we used the *w1118* fly line, and we found an interaction of Ubx with Pho in an in vivo context, which to our knowledge has so far not been shown for a Pho-TF interaction. And actually, we do not propose a direct recruitment or co-recruitment but we state that Ubx stabilizes Pho binding to repressed regions. The weak signal suggests that the interaction of Pho with Ubx is rather weak but essential for the maintenance of the repression. We have now stated this even clearer in the text.

GAGA-like motifs are often observed by motif discovery algorithms and I am not convinced they are meaningful here given the lack of solid follow up evidence.

We assume a lot of genomic regions to contain the GAGA or GAGA-like motif and we agree that motif search does not that these regions are functional or bound by GAGA. However, we do think that some Ubx bound regions will have GAGA bound, as we could show an interaction between Ubx and GAGA at least in cell culture (Co-IP, now Figure 8—figure supplement 1D).

One potential strength of the paper is the claim that Pho binding and K27me3 depends on Ubx binding. However, this conclusion is based on a small handful of loci analyzed by qPCR and the differences are small. Further, given that the Pc system can have very specific targets, I think that calculating the ratio of K27me3/K27ac can be misleading. Better to treat these marks independently, since when one is lower it isn't necessarily the case that the other will be higher. It is also the case that Pho binding is not equivalent to K27me3; the latter could simply be a consequence of whether the gene is repressed, which may or may not occur in a Pho-dependent manner.

We agree with the reviewer and we have redone the analysis using the Pho ChIP-seq dataset we have generated (in control and Ubx degradation backgrounds) to show the global relevance of our finding. In addition, we show K27me3 and K27ac marks independently. This analysis now reveals that a subset of Ubx-Pho bound sites that lose Pho binding in the absence of Ubx have their K27me3 marks significantly reduced, while K27ac marks are increased, demonstrating that at least at a subset of sites Ubx is required for Pho binding and H3K27me3 marks. We have now emphasized this point also stronger in the paper.

Many genes lose expression in the Ubx knockdown experiments, suggesting that Ubx also acts as an activator in the mesoderm, but this is not adequately discussed or studied in the paper. It is not clear why the authors choose to focus on the repressed genes.

We agree with the reviewer that it has not been explained well enough why we chose to focus on repressed genes, which we have now done in the revised version. We have now dedicated also a paragraph in the Discussion on the activation of lineage-specific genes by Ubx to highlight that Ubx controls lineage development not only by repression.

2) General comments.Ubx, and Hox genes in general, are well studied transcription factors that are known to be important in multiple cell types including mesoderm, the nervous system, and epithelium. So, seeing lots of genes change their expression when a broadly used transcription factor like Ubx is knocked down. It is also not surprising that the genes differ depending on the tissue (thus the GO analyses done in this paper seems rather trivial). What would take the analyses beyond what is known already is to understand how Ubx carries out differential functions in these two (and other) tissues. Presumably this happens in conjunction with tissue specific factors (e.g. twist for the mesoderm). The role of lineage specific factors, however, is not addressed and only marginally mentioned in the paper.

We would like to highlight that although it might seem trivial and expected to see a lot of genes changing their expression when Ubx is lost, this is the first time that a Hox TFs, which is active in many different cell types, is studied in individual lineages in vivo. Reviewer #1 and #2 appreciated this aspect a lot, as they thought this is an elegant study. It was in particular this set-up that allowed us to uncover the function of Ubx in repressing alternative fates. We do agree with the reviewer that the next interesting and important question to solve is how Ubx performs this function lineage-specifically, very likely via the interaction with lineage-specific factors. We work on that aspect, but this is beyond the scope of this paper, as this manuscript is already quite complex (stated by reviewer #1).

Terms such as "Cell plasticity" and "lineage restriction" are used rather loosely. The authors show that many up-regulated genes after Ubx knockdown are expressed in other lineages, but this result would have to be the case when a TF that represses genes is removed. There is no hard-evidence of lineage conversion (like loss of mesodermal master-regulatory factors for example), nor is it clear why neuronal (as opposed to endodermal or epithelial genes) genes are the focus here. It seems that the analysis is artificially restricted to repressed neuronal genes, while a more unbiased analysis of the changes that occur would be more valuable.

We went critically through the manuscript and used the terms “cell plasticity” and “lineage restriction” in a more stringent manner. Concerning the second point, that there is no hard evidence for lineage conversion, we would like to highlight that we never state that the mesodermal lineage changes identity in the absence of Ubx. Actually, we dedicated a whole paragraph in the Discussion why we believe that there is no lineage conversion, one reason being that other Hox TFs (like Antp) are ectopically expressed when Ubx is removed, which are very likely involved in maintaining the expression of mesodermal master-regulators. Thus, it will be important to study lineage development in a Hox free environment. However, this is beyond the scope of the paper. We now state more clearly why we focused on neuronal genes in our GO term analysis. However, we would like to highlight that we show the global significance of the de-repression of different types of alternative fate genes in Figure 3O.